# Ammonia pools in zeolites for direct fabrication of catalytic centers

Jie Yao[1,6], Yingluo He [1,6], Yan Zeng[1,6], Xiaobo Feng[1,2], Jiaqi Fan[1], Shoya Komiyama[1], Xiaojing Yong[3], Wei Zhang[3], Tiejian Zhao[3], Zhongshan Guo[3], Xiaobo Peng [1,4✉], Guohui Yang [1,5✉] & Noritatsu Tsubaki [1✉]

Reduction process is a key step to fabricate metal-zeolite catalysts in catalytic synthesis. However, because of the strong interaction force, metal oxides in zeolites are very difficult to be reduced. Existing reduction technologies are always energy-intensive, and inevitably cause the agglomeration of metallic particles in metal-zeolite catalysts or destroy zeolite structure in severe cases. Herein, we disclose that zeolites after ion exchange of ammonium have an interesting and unexpected self-reducing feature. It can accurately control the reduction of metal-zeolite catalysts, via in situ ammonia production from 'ammonia pools', meanwhile, restrains the growth of the size of metals. Such new and reliable ammonia pool effect is not influenced by topological structures of zeolites, and works well on reducible metals. The ammonia pool effect is ultimately attributed to an atmosphere-confined self-regulation mechanism. This methodology will significantly promote the fabrication for metal-zeolite catalysts, and further facilitate design and development of low-cost and high-activity catalysts.

[1] Department of Applied Chemistry, School of Engineering, University of Toyama, Gofuku 3190, Toyama 930-8555, Japan. [2] Jiangsu Province Engineering Research Center of Fine Utilization of Carbon Resources, China University of Mining & Technology, Xuzhou 221116 Jiangsu, China. [3] National Energy Group Ningxia Coal Industry Co., Ltd., No. 168 Beijing Middle Road, Yinchuan, China. [4] National Engineering Research Center of Chemical Fertilizer Catalyst, Fuzhou University, Fuzhou 350002 Fujian, China. [5] State Key Laboratory of Coal Conversion, Institute of Coal Chemistry, Chinese Academy of Sciences, Shanxi 030001 Taiyuan, China. [6] These authors contributed equally: Jie Yao, Yingluo He, Yan Zeng. ✉email: PENG.Xiaobo@fzu.edu.cn; thomas@eng.u-toyama.ac.jp; tsubaki@eng.u-toyama.ac.jp

Since the first synthetic preparation of zeolites by Barrer in the 1940's, zeolites as microporous materials represent a "New Frontier" of solid-state chemistry[1]. Up to now, several kinds of zeolites have been applied as catalysts in the oil refining and petrochemical processes benefitting from their superior thermostability, renewability, and catalytic selectivity[2]. Coordinating with metals, the role of zeolites in industrial catalysis is being further enhanced. Zeolite-supported transition metal catalysts, as a class of bifunctional heterogeneous catalysts, have been successfully applied in industrial catalytic processes, such as methanol to aromatics[3,4], methanol to olefins[5,6], ammonia selective catalytic reduction[7–9], and so on.

Nevertheless, the fabrication process of metal-zeolite catalysts always involves reducing metal precursors under the reducing gas (mainly including $H_2$, CO, or syngas (CO + $H_2$)) flow, which is very energy-intensive. In addition, compared with bulk metals, very fine metal oxides in zeolites are much more difficult to be reduced because of the strong interaction force upon metal-zeolite interface[10–12]. Hence, a higher reduction temperature is usually needed, which inevitably gives rise to the aggregation of metallic particles, and severely weakens the activity and stability of catalysts[13–15]. To date, the fabrication of robust metal-zeolite catalysts with simple, safe, and energy-efficient reduction technology has remained a challenge.

Zeolites in metal-zeolite catalysts, in most cases, provide the acid-catalytic functional sites for reactions[16–18]. However, by now only few zeolites can be synthesized directly as H-type. Most zeolites need to first yield the metal cationic type, the most common case as Na-type, and then transform to H-type by $NH_4^+$ or $H^+$ exchange. In these preparations, ammonium salts as weakly acidic chemicals are widely used to obtain H-type zeolites via ion exchange, then calcination[19–21], because common strong acids usually cause dealumination and structural damage to zeolites[22–24]. It is worth mentioning that in the transformation process of $NH_4$-zeolites to H-zeolites, ammonia molecules will be produced. It is well-known that ammonia is a reducing gas and can reduce metal oxides like CuO readily at a high temperature. However, to the best of our knowledge, there is no record of reducing supported metal species by released ammonia molecules from $NH_4$-zeolites.

Here, we employ a series of zeolites as supports whose pore channels brim with ammonium ions (named 'ammonia pools'), and disclose that these zeolites enable in situ reduction of supported metal species without using any additional reduction process. We find that the ammonia molecules can be continuously released from ammonia pools during the calcination process of $NH_4$-zeolites. Then, these ammonia molecules mildly and efficiently reduce supported metal oxides, while maintaining small sizes and high degree of dispersion for the metallic particles. Such special self-reducing behavior occurring in $NH_4$-zeolites is called 'ammonia pool effect' (abbreviated as APE). Typical catalytic experiments (including carbonylation of dimethyl ether, synthesis of ethylene by methane coupling, and methanol synthesis from methane oxidation) prove that the metal-zeolite catalysts, fabricated via APE, realize higher catalytic activity and selectivity than those prepared by traditional reduction method (Fig. 1). Remarkably, APE is unaffected by topological structures of zeolites, and works well on various metals. An atmosphere-confined self-regulation mechanism is finally proposed to give a more in-depth insight into this interesting phenomenon and effect.

## Results and discussion
### Ammonia pool effect (APE) of Cu-MOR catalysts. 
We employed traditional $H_2$ reduction and APE reduction to fabricate Cu-MOR catalysts, respectively (see Supplementary Fig. 1 for the preparation processes). To confirm the APE for the Cu-MOR samples, we designed a direct detection apparatus to analyze the effluent gas of APE reduction over an $NH_4$–Cu(3.41 wt%)-MOR sample (Supplementary Fig. 2). In the analysis, the $NH_4$–Cu(3.41 wt%)-MOR sample of 10 g was loaded in a reaction tube. Helium gas was used as the sweeping gas, and the sample was reduced by APE at 500 °C for 2 h. Then, the effluent gas was analyzed by an online gas chromatograph (GC) with a thermal conductivity detector (TCD). The result of GC analysis was displayed in Supplementary Fig. 3, and only $N_2$ was detected. To further obtain the temperatures of $N_2$ formation during APE reduction process, we analyzed the $NH_4$–Cu(3.41 wt%)-MOR sample by a system of thermogravimetry-differential thermal analysis and mass spectrometry (TG-DTA-MS). As in Supplementary Fig. 4, the $N_2$ and exothermic peak were detected at about 400 °C. It indicates that ammonia was oxidized into $N_2$ molecule, and APE reduction was realized on the copper species. We also proposed the possible reactions of different copper species in the APE process, as follows:

$$NH_4 - Z \rightarrow NH_3 + H - Z \tag{1}$$

$$6Cu^{2+} + 2NH_3 \rightarrow 6Cu^+ + 6H^+ + N_2 \tag{2}$$

$$6Cu^+ + 2NH_3 \rightarrow 6Cu^0 + 6H^+ + N_2 \tag{3}$$

$$Cu^{2+} + H_2O \rightarrow CuO + 2H^+ \tag{4}$$

$$6CuO + 2NH_3 \rightarrow 3Cu_2O + 3H_2O + N_2 \tag{5}$$

$$3Cu_2O + 2NH_3 \rightarrow 6Cu^0 + 3H_2O + N_2 \tag{6}$$

$$3CuO + 2NH_3 \rightarrow 3Cu^0 + 3H_2O + N_2 \tag{7}$$

Note: In Eq. (1), Z represents the zeolites. Equations (2) to (7) all involve the reduction process of different copper species. The $N_2$ was detected, suggesting oxidation of ammonia molecules and reduction of copper species.

We further prepared the Cu-MOR samples with Cu contents from 1.14 to 5.72 wt% (denoted as $NH_4$–Cu(X)-MOR). The In-situ monitoring of APE process over the $NH_4$–Cu(X)-MOR samples was carried out via differential thermal analysis (DTA) in air atmosphere. In Fig. 2a, the $NH_4$-MOR had a broad endothermic peak at the range of 275–490 °C on the DTA curve. This peak was attributed to the decomposition of $NH_4^+$ to $NH_3$ and $H^+$. Correspondingly, the $NH_4$–Cu(X)-MOR samples possessed narrow exothermic peaks at 300–450 °C, which exactly located in the temperature range of decomposition of $NH_4^+$ to $NH_3$. Our TG-DTA-MS analysis also demonstrated the $N_2$ formation at that temperature range. Hence, these exothermic peaks should be ascribed to the reduction of copper species by $NH_3$. Interestingly, when more $Cu^{2+}$ ions were anchored in $NH_4$-MOR zeolites, the centers of exothermic peaks shifted to higher temperatures. This phenomenon indicates that a higher degree of $Cu^{2+}$ exchange would increase the difficulty of APE reduction.

All $NH_4$–Cu(X)-MOR samples displayed an endothermic peak below 100 °C, which corresponded desorption of physically adsorbed $H_2O$. Compared to the $NH_4$-MOR, the $NH_4$–Cu(X)-MOR samples showed sharper increases for the DTA profiles in the temperature range of 100–275 °C, due to partial reaction of $Cu^{2+}$ and $H_2O$ to CuO and $H^+$. At the higher temperature stage of 485–595 °C, the unobvious broad exothermic peaks over the $NH_4$–Cu(X)-MOR samples were attributed to the partial oxidation of cuprous oxide and metallic copper, because of the air atmosphere of the DTA experiment. Therefore, the working temperature range for APE on Cu-MOR samples was 300–450 °C.

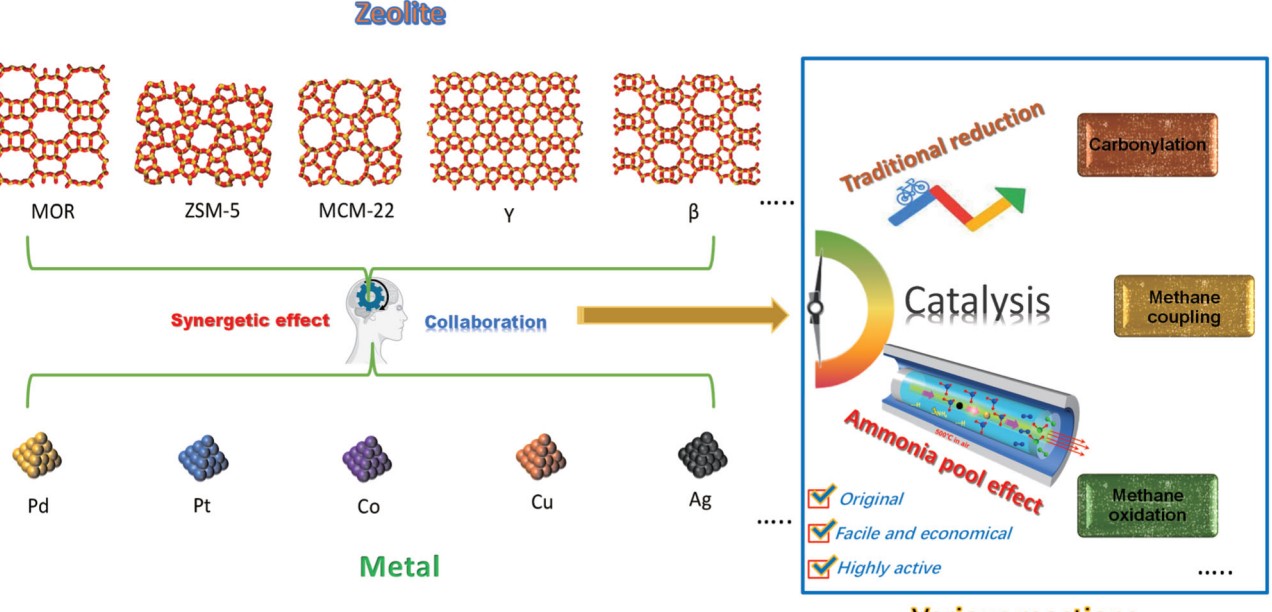

**Fig. 1 The ammonia pool effect in different types of zeolites and metals for various catalytic reactions.** Metal-zeolite catalysts, consisting of many base/noble metals (Fe, Co, Ni, Cu, Pt, Ag, Au, and Pb) being incorporated in varied zeolites (as MOR, ZSM-5, MCM-22, Y, β), are rationally fabricated by APE to realize considerably better synergetic effect in a series of reactions involving carbonylation, methane coupling, and methane oxidation. The presented APE is original, facile, economical, and highly efficient.

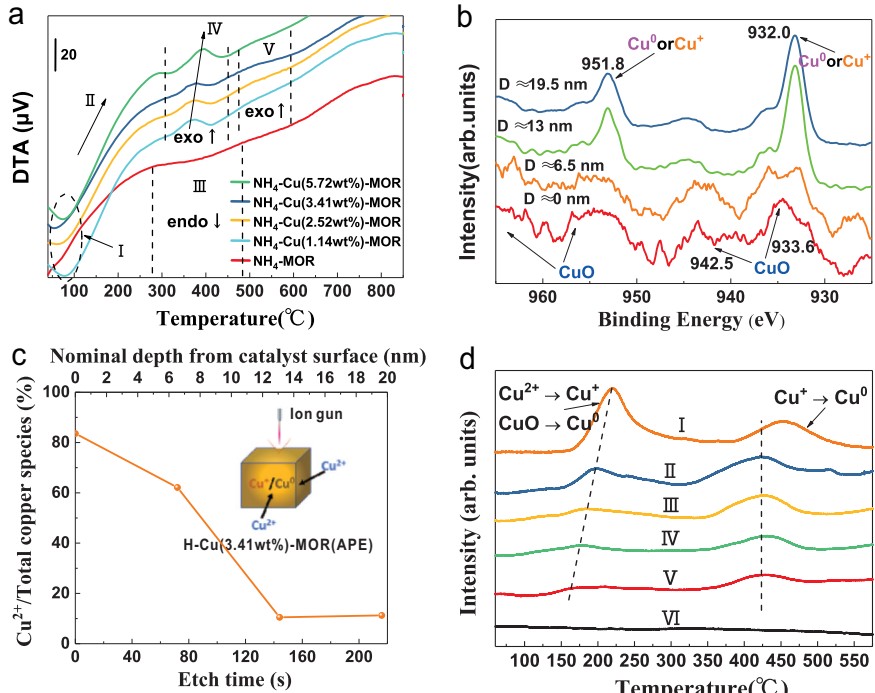

**Fig. 2 In-depth understanding of APE. a** DTA results of NH$_4$-MOR and NH$_4$-Cu(X)-MOR samples with different copper contents. Note: I: desorption of physically adsorbed H$_2$O; II: H$_2$O + Cu$^{2+}$→CuO + 2H$^+$; III: NH$_4$-MOR → H-MOR + NH$_3$; IV: NH$_3$ + Cu$^{2+}$/CuO→Cu$^+$ + Cu$^0$ + N$_2$ + H$^+$; V: Cu$^+$/Cu$_2$O → CuO or Cu$^0$ → Cu$_2$O/CuO. The exothermic and endothermic behaviors are abbreviated as exo and endo, respectively. **b** XPS depth profile of H-Cu(3.41 wt%)-MOR(APE). D represents the etching depth. **c** The relationship between Cu$^{2+}$ proportion and nominal depth from the catalyst surface. **d** The H$_2$-TPR profiles of different Cu-MOR samples. Note: I: H-Cu(3.27 wt%)-MOR(Air); II: H-Cu(5.72 wt%)-MOR(APE); III: H-Cu(3.41 wt%)-MOR(APE); IV: H-Cu(2.52 wt%)-MOR(APE); V: H-Cu(1.14 wt%)-MOR(APE); VI: H-Cu(3.27 wt%)-MOR(H$_2$). The H-Cu(X)-MOR(APE) samples, with decreasing Cu contents, showed lower reduction temperature for the low-temperature peaks, indicating easy reduction of the copper species.

Once the temperature rose to higher than 500 °C, oxidation of as-reduced copper species took place. An optimal APE should be controlled below 500 °C in air.

To obtain the composition and proportion of different copper species in samples fabricated via APE, an argon ion etching assisted X-ray photoelectron spectra (XPS) was employed on a Cu-MOR sample after APE (H–Cu(3.41 wt%)-MOR(APE)). The distribution of copper species along the depth direction on sample was shown in Fig. 2b. It was clear that copper oxide was the main copper species on the surface of H–Cu(3.41 wt%)-MOR(APE), and the content of reduced copper species increased sharply when the depth enlarged (Fig. 2c). The content of copper oxide in total copper species maintained at about 10% when the nominal depth was larger than 13 nm. This interesting phenomenon should be attributed to the slight oxidation of metal from the outside to inside of zeolite by air during calcination.

The Cu 2p XPS spectra of Cu-MOR samples (H–Cu(X)-MOR(APE)) with different copper contents after APE and etching treatment were compared in Supplementary Fig. 5a. The characteristic peak of CuO became more obvious when there was higher copper content in samples. Because the total $NH_4^+$ amount in $NH_4$-MOR was constant, more ion exchange of $Cu^{2+}$ with $NH_4^+$ led to less $NH_4^+$ left in $NH_4$–Cu(X)-MOR sample. In the course of APE, the amount of $NH_3$ produced from the decomposition of $NH_4$–Cu(X)-MOR to H–Cu(X)-MOR was positively related to the amount of $NH_4^+$ in $NH_4$–Cu(X)-MOR. Therefore, after more exchanges of $Cu^{2+}$ for the $NH_4$-MOR, the $NH_3$ molecules were insufficient for the reduction of copper oxide. According to these analyses, we confirm that moderate ion-exchange of $Cu^{2+}$ is a key factor of APE reduction in the $NH_4$–Cu(X)-MOR samples. To further distinguish between the $Cu^+$ intermediate and $Cu^0$ species, we employed Auger electron spectroscopy (AES)[25,26] to characterize the H–Cu(X)-MOR(APE) samples (Supplementary Fig. 5b, c). The Cu LMM Auger spectra clearly revealed that high proportion of $Cu^+$ intermediate was formed on the H-Cu(X)-MOR(APE) samples after the APE reduction.

Quantitative analysis of different copper species over the H–Cu(X)-MOR(APE) samples was achieved by temperature-programmed reduction of $H_2$ ($H_2$-TPR) measurement. H–Cu(3.27 wt%)-MOR(Air) and H–Cu(3.27 wt%)-MOR($H_2$) were prepared from $Cu^{2+}$ ion-exchange of H-MOR zeolite, and then calcined by air and reduced through $H_2$, respectively. These two samples were also measured by the $H_2$-TPR instrument. The $H_2$ consumption curves of the H–Cu(X)-MOR(APE), H–Cu(3.27 wt%)-MOR(Air), and H–Cu(3.27 wt%)-MOR($H_2$) were displayed in Fig. 2d, and the corresponding peaks to different copper species were labeled based on the related literatures[27–30]. The H–Cu(3.27 wt%)-MOR($H_2$) did not show any characteristic peak, because the copper species have been completely reduced. Compared with the H–Cu(3.27 wt%)-MOR(Air), all the H–Cu(X)-MOR(APE) exhibited lower temperature and intensity of $H_2$ consumption peaks. It indicates that the copper species of the H–Cu(X)-MOR(APE) were easier to be reduced, than that of the H–Cu(3.27 wt%)-MOR(Air). Moreover, they have been partially reduced during the APE process. The percentages of different copper species were calculated and displayed in Supplementary Table 1. The total amount of Brønsted acid sites, obtained from the TPD of $NH_4$-MOR (Supplementary Fig. 6), was used to determine the ion-exchange degree of copper. The results revealed that the proportion of $Cu^0$ in the H–Cu(X)-MOR(APE) increased with decreasing the copper exchange degree (see Supplementary Table 1).

To confirm the stability of crystal structures, traditional X-ray diffraction (XRD) analysis was conducted on the $NH_4$–Cu(X)-MOR and H–Cu(X)-MOR(APE), respectively. Both the $NH_4$–Cu(X)-MOR and H–Cu(X)-MOR(APE) did not display any changes of the XRD patterns, implying the stable structures

(Supplementary Fig. 7a, b). No peak associated with metal species was observed in these samples, indicating that metal species were highly dispersed in MOR zeolites. In situ XRD was further employed to reveal the crystal features of $NH_4$–Cu(X)-MOR during APE reduction. As in Supplementary Fig. 8, the $NH_4$–Cu(3.41 wt%)-MOR, as a reference sample, was heated to 500 °C, and the XRD patterns were recorded in the temperature range from 300 to 500 °C. The In situ XRD patterns at each temperature were the same. The results further demonstrated that the APE reduction did not injure the crystal structures of MOR zeolite.

Transmission electron microscopy (TEM) was utilized to further probe morphological features for H–Cu(X)-MOR(APE) and H–Cu(3.27 wt%)-MOR($H_2$) samples, and the results were listed in Supplementary Fig. 9. The H–Cu(1.14 wt%)-MOR(APE) did not show few obvious copper nanoparticles, implying high dispersion of the copper species (Supplementary Fig. 9a). Further increasing copper content in the samples, the H–Cu(2.52 wt%)-MOR(APE), H–Cu(3.41 wt%)-MOR(APE), and H–Cu(5.72 wt%)-MOR(APE) exhibited well-dispersed copper particles in the MOR zeolite, and the average size of copper particles increased gradually (Supplementary Fig. 9b, c, and e). Notably, the TEM image of H–Cu(3.27 wt%)-MOR($H_2$) uncovered that the copper particles were generally located on the surface of the MOR zeolite (Supplementary Fig. 9d), and the average size of copper particles was much larger than that of the H–Cu(3.41 wt%)-MOR(APE). We also utilized in situ TEM to observe the precursor of H–Cu(3.27 wt%)-MOR($H_2$) in air calcination (Supplementary Fig. 10). The in-situ process revealed rapid agglomeration of the copper species on the surface. According to these TEM and in situ TEM observations, we concluded that the traditional method of air calcination and $H_2$ reduction can easily lead to the agglomeration of copper particles, but APE reduction is a successful way to inhibit it.

In addition, our scanning electron microscope (SEM) and energy-dispersive X-ray spectroscopy (EDS) analyses showed that the elemental mapping of Cu on the H–Cu(3.41 wt%)-MOR(-APE) was weaker than that on the H–Cu(3.27 wt%)-MOR($H_2$) under the same analysis conditions (Supplementary Fig. 11). This illustrates that the H–Cu(3.41 wt%)-MOR(APE) possessed less Cu species on the surface, and more Cu species in the matrix of MOR zeolite, compared with the H–Cu(3.27 wt%)-MOR($H_2$). This finding is consistent with the results of TEM in Supplementary Fig. 9. Further, $N_2$ adsorption-desorption isotherms of the H–Cu(X)-MOR(APE) displayed a combination of type I and IV isotherms (Supplementary Fig. 12), which implies the existence of micropore and mesopore structure[31]. In Supplementary Table 2, the textural parameters unveiled a declined BET surface area, when the copper content increased in the H–Cu(X)-MOR(APE). It should be ascribed to the gradual plugging of copper particles to the channels of MOR zeolites. Although the BET surface area decreased over the H–Cu(X)-MOR(APE), the external surface area, micropore volume, and pore size did not change significantly. It indicates that MOR zeolite still possessed a good porosity for catalytic reaction.

Because acid property is an important feature of zeolite catalysts, we characterized the H–Cu(X)-MOR(APE) via temperature-programmed desorption of $NH_3$ ($NH_3$-TPD). As in Supplementary Fig. 13, the $NH_3$-TPD profiles can be mainly divided into two regions[32]. The low-temperature peak at around 200 °C was due to physical adsorption of $NH_3$ molecules, along with weak interaction between $NH_3$ molecules and weak acid sites. The high-temperature peak at around 550 °C was mainly attributed to $NH_3$ molecules adsorbed on Brønsted acid sites. In addition, with the increase of copper content, the peak intensity decreased both in the high-temperature and low-temperature

regions. This implies that ion exchange of $Cu^{2+}$ can weaken the acid properties[33]. The shoulder peak at around 325 °C on H–Cu(5.72 wt%)-MOR(APE) should be originated from desorption of $NH_3$ molecules from medium acid sites, owing to high degree of $Cu^{2+}$ exchange[34]. According to the $NH_3$-TPD, TEM, and $N_2$ physisorption analyses, we consider that the moderate content of copper was not only beneficial to keep acid properties of the Cu-MOR catalysts, but also in favor of maintaining the high degree of copper dispersion and zeolite porosity, after the APE reduction.

**APE of transitional or noble metal-zeolite catalysts.** In addition to the Cu-MOR catalysts, we further fabricated transitional metal-MOR catalysts and noble metal-MOR catalysts, to explore the features of APE reduction. The transitional metals (Fe, Co, and Ni) and the noble metals (Pt and Au) were utilized to synthesize the Fe-MOR, Co-MOR, Ni-MOR, Pt-MOR, and Au-MOR samples with the metal contents of 0.9–1.8 wt%. The traditional $H_2$ reduction and APE reduction were also employed to synthesize these samples, respectively (see Supplementary Fig. 1 for the preparation processes). The XPS results revealed that, after APE reduction, all the transitional metal-MOR catalysts and noble metal-MOR catalysts (i.e., Fe(1.8 wt%)-MOR(APE), Co(0.9 wt%)-MOR(APE), Ni(1.2 wt%)-MOR(APE), Pt(0.9 wt%)-MOR(APE) and Au(1.1 wt%)-MOR(APE)), exhibited characteristic peaks of the zerovalent metals (Supplementary Fig. 14a–e). It clearly proved that these metal samples were reduced by the APE process. In particular, the Pt(0.9 wt%)-MOR(APE) and Au(1.1 wt%)-MOR(APE) generated higher proportion of zerovalent peaks, than the other samples. This indicates that the noble metals (Pt and Au) on the MOR were easier to be reduced than the transitional metals (Fe, Co, and Ni).

The Fe-MOR, Co-MOR, Ni-MOR, Pt-MOR, and Au-MOR samples, after traditional hydrogenation reduction and APE reduction, were also investigated by XRD and $H_2$-TPR. As shown in Supplementary Fig. 15, the XRD results uncovered that all the samples still possessed stable crystal structures for the MOR zeolite. The $H_2$-TPR analysis demonstrated that the reduction degree of the Fe(1.8 wt%)-MOR(APE), Co(0.9 wt%)-MOR(APE), and Ni(1.2 wt%)-MOR(APE) reached 86, 51, and 67%, respectively (Supplementary Fig. 16a–c). Moreover, the reduction degree of the Pt(0.9 wt%)-MOR(APE) and Au(1.1 wt%)-MOR(APE) was as high as about 90% (Supplementary Fig. 16d, e). In control experiment, Au(1.0 wt%)-MOR(Ar) in Supplementary Fig. 16e was synthesized via calcination in Ar atmosphere, and in situ reduced by zeolite dehydration[35–37]. Our TG-TDA and TG-DTA-MS analyses also proved that the MOR zeolite can be dehydrated at the temperature range of 100–250 °C (Fig. 2a and Supplementary Fig. 17). However, the reduction degree of the Au(1.0 wt%)-MOR(Ar) was only 31 %, and much lower than that of the Au(1.1 wt%)-MOR(APE).

In view of the high degree of APE reduction on the Pt-MOR and Au-MOR samples, we further utilized noble metals of Pd and Ag, and different types of zeolites (ZSM5, Y, β, and MCM-22), to prepare the noble metal-zeolite catalysts. In $H_2$-TPR analysis, Ag(2.3 wt%)-ZSM-5($H_2$), Pd(1.2 wt%)-Y($H_2$), Pd(1.3 wt%)-β($H_2$), and Pd(1.2 wt%)-MCM-22($H_2$) samples, prepared by traditional $H_2$ reduction, did not exhibit obvious peaks of $H_2$ consumption (Supplementary Fig. 18a–d). Ag(2.4 wt%)-ZSM-5(APE), Pd(1.4 wt%)-Y(APE), Pd(1.3 wt%)-β(APE), and Pd(1.3 wt%)-MCM-22(APE) samples, obtained from APE reduction, displayed lower intensity of $H_2$-TPR peaks, than Ag(2.3 wt%)-ZSM-5(Air), Pd(1.2 wt%)-Y(Air), Pd(1.3 wt%)-β(Air), and Pd(1.2 wt%)-MCM-22(Air) samples synthesized from air calcination. These phenomena are in good agreement with the $H_2$-TPR results of the Pt-MOR,

Au-MOR, and transitional metal-MOR samples. Ag(2.4 wt%)-ZSM-5(E-$NH_3$) was fabricated by external $NH_3$ reduction, and did not visibly display the $H_2$-TPR peaks (Supplementary Fig. 18a). It directly demonstrated that $NH_3$ molecule is effective to reduce metal-zeolite catalysts.

TEM characterization was also employed to analyze the Ag-ZSM-5, Pd-β, and Pt-MOR samples after traditional $H_2$ reduction and APE reduction (Supplementary Figs. 19–21). The Ag(2.3 wt%)-ZSM5($H_2$), Pd(1.3 wt%)-β($H_2$), and Pt(0.9 wt%)-MOR($H_2$) samples after traditional $H_2$ reduction exhibited agglomeration of metal particles on the surface (Supplementary Figs. 19b, 20b, and 21b). But the Ag(2.4 wt%)-ZSM5(APE), Pd(1.3 wt%)-β(APE), and Pt(0.9 wt%)-MOR(APE) after APE reduction successfully inhibited the agglomeration, and improved the dispersity of metal species (Supplementary Figs. 19a, 20a, and 21a). The results were the same as those of the Cu-MOR samples (Supplementary Fig. 9c, d). SEM and EDS analyses on the Ag(2.4 wt%)-ZSM5(APE), Ag(2.3 wt%)-ZSM5($H_2$), Pd(1.4 wt%)-Y(APE), and Pd(1.2 wt%)-Y($H_2$) were displayed in Supplementary Fig. 22 and 23. The observed phenomena also revealed that the APE reduction led to less metal species on the surface, than the traditional $H_2$ reduction. In addition, $N_2$ physisorption was conducted on the Ag-ZSM5 and Pd-Y samples after traditional $H_2$ reduction and APE reduction (Supplementary Table 3). The physisorption analysis uncovered that the noble metal-zeolite catalysts with low metal contents still kept high BET areas. Based on these experimental analyses, we confirm that APE reduction can be widely used both in transitional metal-zeolite catalysts and noble metal-zeolite catalysts.

**Superior catalytic performance.** To obtain the physicochemical properties, the transitional metal-zeolite catalysts and noble metal-zeolite catalysts have been analyzed by multiple characterization techniques. However, it is also important to gain their catalytic properties. Therefore, we further evaluated the typical metal-zeolite catalysts, in carbonylation of dimethyl ether (DME), synthesis of ethylene by methane coupling, and methanol synthesis from methane oxidation, respectively. The catalytic results were shown in Fig. 3 and Supplementary Tables 4–6.

DME carbonylation is a core reaction in the new ethanol synthetic route proposed in our previous work[38–43]. For zeolite-catalyzed DME carbonylation, it is gaining acceptance that reduced copper species can effectively promote the activity of zeolite catalysts[44–47]. As in Supplementary Table 4, MOR zeolites were modified by copper species with different valence states. The metal copper ($Cu^0$), via a traditional $H_2$ reduction, behaved the best catalytic performance. Oppositely, the monovalent and divalent copper species inhibited the carbonylation. Methanol as by-product was largely generated due to the hydration reaction of DME and $H_2O$[48]. This result agrees well with the reported conclusion and proves that $Cu^0$ is a good promoter in zeolite-catalyzed DME carbonylation[45,46]. However, it is surprising that all the H–Cu(X)-MOR(APE) catalysts displayed higher carbonylation performance than the H–Cu(X)-MOR($H_2$) catalysts (Fig. 3a and Supplementary Table 5), although the $Cu^0$ content of H–Cu(X)-MOR(APE) was lower than that of H–Cu(X)-MOR($H_2$) (Fig. 2d and Supplementary Table 1). In addition, the H–Cu(3.41 wt%)-MOR(APE), with the highest ratio of $Cu^+$ (40.6%) to $Cu^0$ (45.8%) (See Supplementary Table 1), generated the best performance of DME carbonylation. Hence, we confirm that the coexistence and synergy of $Cu^0$ and $Cu^+$ played the key role in enhancing the DME carbonylation of H-Cu(X)-MOR(APE).

Note that the TG results in Supplementary Fig. 24 showed that the used H–Cu(3.41 wt%)-MOR(APE) had less amount of coke, than the spent H–Cu(3.27 wt%)-MOR($H_2$). This suggests that a

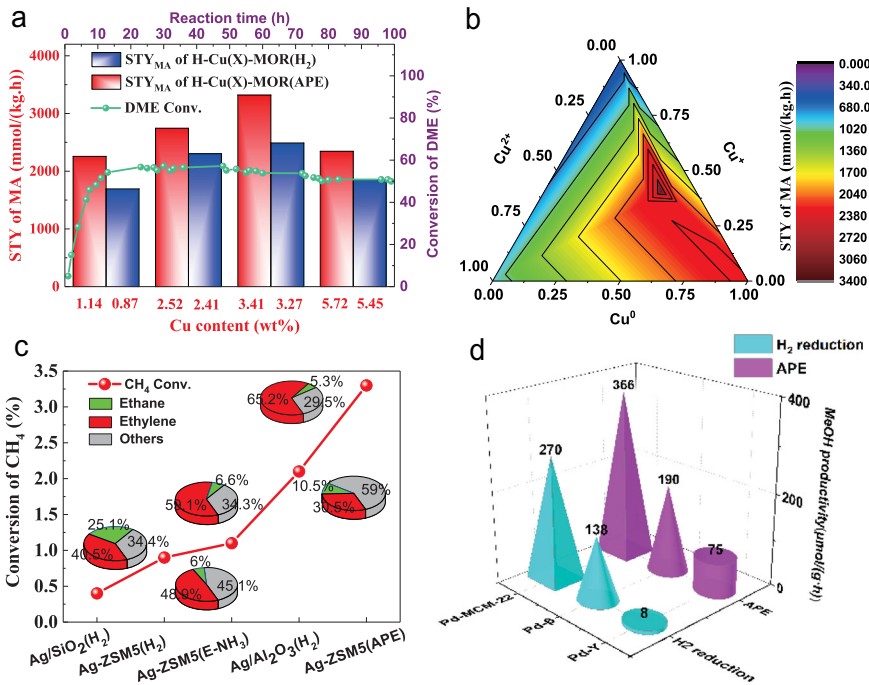

**Fig. 3 Catalytic performance. a** The catalytic activity comparison of Cu-MOR catalysts reduced by APE and $H_2$ in DME carbonylation, and the stability test of Py-Cu(3.41 wt%)-MOR(APE). Reaction condition: 0.5 g catalyst, reaction temperature 220 °C, reaction pressure 1.5 MPa, reaction time 8 h, reaction gas DME/CO/Ar = 4.1/92.8/3.1 (vol %), flow rate = 20 mL/min, GHSV (Gas Hour Space Velocity) = 2400 mL (g h)$^{-1}$. Stability test was accomplished under 4 MPa, reaction gas DME/CO/$H_2$/Ar = 5.0/31.9%/60.0%/3.1 (vol %). **b** A predictive performance model for DME carbonylation on the H-Cu(X)-MOR(APE) catalysts. It was based on the distribution of copper valences, copper content, and STY of MA. **c** The catalytic performance of different Ag-based catalysts in synthesis of ethylene by methane coupling. The pie charts represent the product selectivity. The selectivity of the other products included heavy hydrocarbons of benzene, toluene, naphthalene, and coke. The Ag-ZSM5(E-$NH_3$) was reduced by externally introduced $NH_3$ gas. Reaction condition: 0.5 g catalyst, reaction temperature 800 °C, reaction pressure 0.1 MPa, reaction gas $CH_4$/Ar = 90.3%/9.7% (vol %), flow rate = 12.5 mL/min, GHSV (Gas Hour Space Velocity) = 1500 mL (g h)$^{-1}$. **d** The catalytic performance of methanol synthesis from methane oxidation over different Pd-zeolite catalysts after reduction of APE or $H_2$. Reaction condition: 10 mg catalyst, 0.5 M $H_2O_2$ in 10 mL $H_2O$, reaction temperature 70 °C, reaction pressure 3.0 MPa ($CH_4$), reaction time 30 min. On the Pd-Y samples, the catalyst weight was 30 mg, and the reaction temperature was 50 °C.

superior anti-coke ability of the catalyst was produced via APE reduction. After selective poisoning by pyridine[49,50], Py-Cu(3.41 wt%)-MOR(APE) achieved stable catalysis for 100 h and maintained MA selectivity of 94.5% (Fig. 3a and Supplementary Table 5). Further, the TEM images in Supplementary Fig. 25 uncovered that the average sizes of copper particles have no obvious change on the fresh H–Cu(3.41 wt%)-MOR(APE) and the spent Py-Cu(3.41 wt%)-MOR(APE). This implies that a robust agglomeration-resisting nature of the copper species was obtained by APE reduction.

Based on the distribution of copper valences, copper content, and STY of MA, we built a predictive performance model of DME carbonylation on the H–Cu(X)-MOR(APE) catalysts (Fig. 3b). This model suggests that different copper species generate a synergistic catalytic effect to DME carbonylation, and are in agreement with the contribution order of first $Cu^0$, next $Cu^+$, and last $Cu^{2+}$. This finding will provide help for the rational design of copper-modified MOR catalysts in DME carbonylation.

Direct synthesis of ethylene by methane coupling is a very promising reaction, which can convert natural gas into high added-value chemicals[51]. We selected this reaction to further verify the superiority of APE technology for the noble metal-zeolite catalysts, due to its high reaction temperature (800 °C). In Fig. 3c, we evaluated Ag-based catalysts loaded on different supports (containing about 2.4 wt% Ag; see "Methods" section for details) for the reaction. In the terms of conversion of methane, it was in the order of Ag-ZSM5(APE) > Ag/$Al_2O_3$($H_2$) > Ag-ZSM5(E-$NH_3$) > Ag-ZSM5($H_2$) > Ag/$SiO_2$($H_2$). Among these

catalysts, Ag-ZSM5(APE), i.e., Ag(2.4 wt%)-ZSM5(APE), exhibited the best coupling ability, and its ethylene selectivity can reach as high as 65.2%. According to these reaction results, we found that the Ag-ZSM5(APE) from APE reduction was better than the other Ag-based catalysts from traditional $H_2$ reduction or external $NH_3$ reduction. Notably, the used Ag-ZSM5(APE) also showed much less coke, than the spent Ag-ZSM5($H_2$) (Supplementary Fig. 26). This observation demonstrated that the superior anti-coking ability was also formed on the Ag-ZSM5(APE) after APE reduction.

In consideration of liquid-phase reaction environment, we chose the reaction of low-temperature direct oxidation of methane into methanol[52], to further evaluate the noble metal-zeolite catalysts. To investigate the zeolite effects during APE reduction, we employed the Pd-zeolite catalysts with three types of zeolite supports (including MCM-22, β, and Y). The resulting Pd-MCM-22, Pd-β, and Pd-Y catalysts, after traditional $H_2$ reduction and APE reduction, were the same as the characterized samples of Pd(1.2 wt%)-MCM-22($H_2$), Pd(1.3 wt%)-MCM-22(APE), Pd(1.3 wt%)-β($H_2$), Pd(1.3 wt%)-β(APE), Pd(1.2 wt%)-Y($H_2$) and Pd(1.4 wt%)-Y(APE). Their catalytic performances were shown in Fig. 3d and Supplementary Table 6. All the Pd-zeolite catalysts after APE reduction showed higher methane conversion. Moreover, they also exhibited higher methanol productivity, than the Pd-zeolite catalysts from traditional $H_2$ reduction. Especially, the methanol productivity of Pd-Y after APE was almost one order of magnitude higher than that of Pd-Y after traditional $H_2$ reduction. These unexpected findings further

demonstrated that the APE reduction could be widely applied in various types of zeolites and reaction environments.

**The atmosphere-confined-self-regulation mechanism.** The aluminosilicate zeolites, in most cases, have the ability of ion exchange with metal cations[53]. After ion exchange, the traditional reduction method is difficult to completely reduce metal cations to metal (<300 °C), because of the strong interaction force between metal ions and zeolites[54]. In addition, reducing gases are hard to access adequately to the metal species inside the deeper zeolite channels or core of metal clusters, owing to the space limitations. However, in the case of APE reduction, metal ions are anchored near to $NH_4^+$ by ion exchange in zeolite frameworks. Due to the neighboring $NH_4^+$, the ion-exchanged metal-zeolite samples, during the calcination process, gradually release ammonia molecules, to effectively reduce the metal species. The comparison of the mechanism of traditional reduction and APE reduction is illustrated in Fig. 4a.

Based on these analyses, we further propose an atmosphere-confined-self-regulation mechanism to explain this APE process. As shown in Fig. 4b, the $NH_3$ from $NH_4^+$ decomposition reacts with the metal precursors and reduces them (Eqs. 1–7). Meanwhile, the $N_2$ product pushes the $H_2O$ molecules out from zeolite channels. Because the channels are filled with the $N_2$ product and continuously released $NH_3$, the outside $O_2$ or $H_2O$ from air atmosphere cannot enter and only slightly oxidize the metal species on outside surface, as proved by our XPS results (Fig. 2b, c). As a result, the APE process is performed smoothly, and high degree of metal dispersion is realized inside the zeolite

(Supplementary Figs. 9 and 19–21). This atmosphere-confined-self-regulation mechanism, in theory, is suitable for various $NH_4$-type zeolites exchanged with different metal ions.

In conclusion, we presented a zeolite-specific self-reducing methodology, i.e., reduction via APE, and proved this widely-applied and convenient technology for efficient catalysis. The features of APE technology were disclosed based on a series of rationally designed characterizations and experiments. We found that the APE technology was not influenced by different structures and types of zeolites, and acted well on various metal ions. Moreover, compared with traditional reduction manners, the superior APE technology fabricated smaller and more dispersed active metal particles in metal-zeolite catalysts. Most importantly, these catalysts further realized higher catalytic activity and selectivity than those obtained via traditional $H_2$ reduction. We finally put forward an atmosphere-confined self-regulation mechanism and clearly explain this interesting but important phenomenon. This work uncovers a hidden specialty of zeolites. We anticipate that the APE strategy demonstrated here will be pivotal for energy-saving fabrication of highly active and stable catalysts, for conversion of important molecules (such as methane, syngas, dimethyl ether, etc.) to high value-added chemicals.

## Methods

**Preparation of catalysts**. The commercial Na-MOR zeolite (Si/Al = 9, Tosho Co., Ltd, Japan) was used as the parent zeolite. The $NH_4$-MOR sample was obtained from the Na-MOR by ion exchange. Na-MOR zeolite was dispersed in $NH_4NO_3$ (Wako chemical, Co., Ltd, Japan) aqueous solution (1 mol/L, the solid of 1 g in 30 mL solution) at 80 °C and ion exchanged for 6 h. The solid was collected by

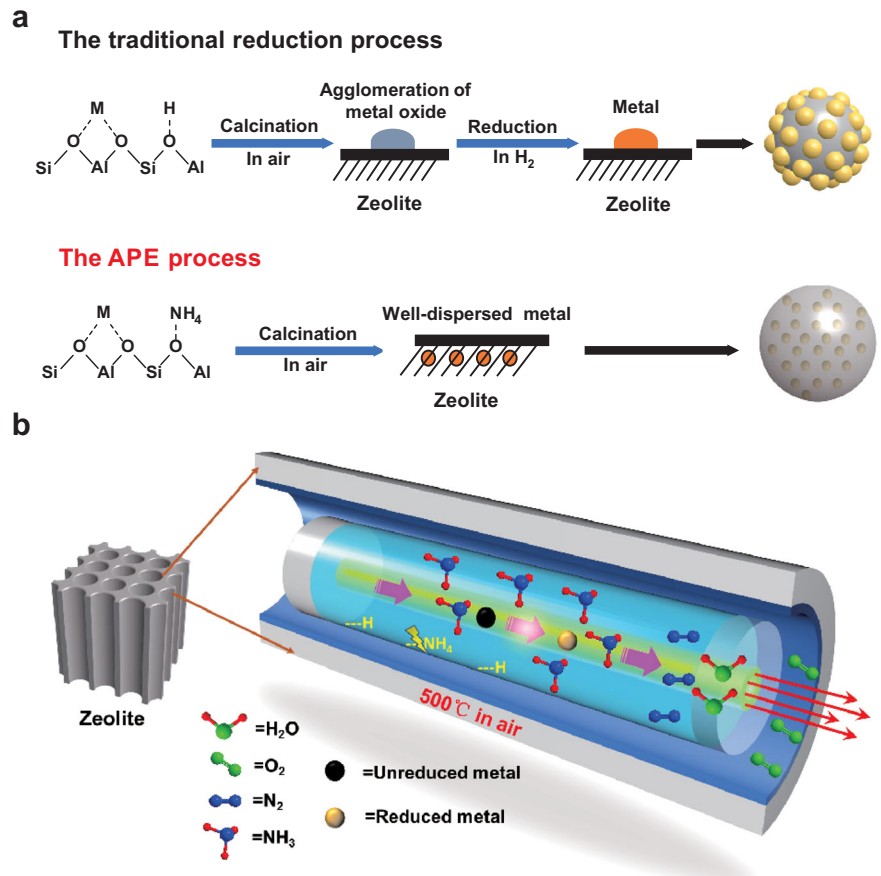

**Fig. 4 The atmosphere-confined-self-regulation mechanism. a** The distinction in mechanism between traditional and APE reduction for metal-zeolite catalysts (M represents the divalent metal ion; the yellow balls and gray ball correspond to metal nanoparticles and zeolite, respectively). **b** Schematic illustrations of the APE process.

filtration and washed with deionized water, then the drying was carried out at 120 °C for 12 h. This process was repeated several times to achieve the complete $NH_4^+$ exchange.

To prepare the copper-modified samples, $NH_4$-MOR was ion exchanged with $Cu(NO_3)_2$ 3$H_2$O (Wako chemical, Co., Ltd, Japan) solution. The process of $Cu^{2+}$ exchange was the same as the description above, except for the concentration of $Cu^{2+}$ aqueous solution (0.1 mol/L). This ion-exchange process can be repeated several times to get the desired copper amount in zeolite. Thereafter, the precursors with different copper contents were calcined at 500 °C for 2 h with a heating rate of 3 °C/min to reduce the copper species with released ammonia molecules in situ, meanwhile, the $NH_4$-type zeolite was transformed to H-type. As-prepared catalysts were named as H–Cu(X)-MOR(APE), where X refers to the mass percent of copper in zeolite. These catalysts were stored under vacuum and used for DME carbonylation directly.

As a comparison, the Cu-MORs samples reduced by $H_2$ were also prepared. The preparation process of $NH_4$-MOR was the same as the description above. After obtaining the $NH_4$-MOR samples, they were calcined at 500 °C for 2 h with a heating rate of 3 °C/min to prepare H-MOR samples. Then the $Cu^{2+}$ exchange was implemented, which was same with the description above. After being calcined at 500 °C for 2 h with a heating rate of 3 °C/min, the pure $H_2$ reduction was conducted at 400 °C for 10 h with 40 mL/min flow rate in a reaction tube. As-prepared catalysts were named as H–Cu(X)-MOR($H_2$). The Cu-MOR catalyst with 3.27 wt% Cu content without $H_2$ reduction is named as H–Cu(3.27 wt%)-MOR(Air). The DME carbonylation was carried out directly when the temperature was cooled to the reaction temperature of 220 °C.

The samples named $Cu^{2+}$-MOR, $Cu^+$-MOR, and CuO-MOR were prepared as follows. The preparation of $Cu^{2+}$-MOR has the same ion-exchange process as introduced above with the H-MOR as the parent catalyst, and this catalyst was obtained by drying at 120 °C for 12 h without the high-temperature calcination; $Cu^+$-MOR was prepared by a solid-state ion-exchange (SSIE) method: a mixture of H-MOR and purified CuCl (Sigma-Aldrich) was exposed to flowing $N_2$ at 60 mL/min, and the temperature was ramped to 500 °C with a heating rate of 3 °C/min, and kept for 3 h; CuO-MOR was prepared by calcination of $Cu^{2+}$-MOR at 500 °C for 2 h with a heating rate of 3 °C/min in air. The copper contents of these three catalysts were controlled in the range of 3–4 wt%.

The catalyst modified by pyridine (Py, Sigma-Aldrich) was prepared as follows: 0.5 g H–Cu(3.41 wt%)-MOR(APE) was loaded in the medium of reaction tube, the temperature was first heated up to 300 °C under following $N_2$ (20 mL/min) and kept for 3 h. Then the saturated Py steam was carried out from a can with 20 mL/min $N_2$ under room temperature into reaction tube. The Py modification was kept for 3 h at 300 °C under following weak $N_2$, then the valve of Py can was closed, pure $N_2$ (50 mL/min) was used to clean the weak adsorbed Py on catalyst for another 3 h at 300 °C.

The preparation processes of Fe-MOR($H_2$), Fe-MOR(APE), Co-MOR($H_2$), Co-MOR(APE), Ni-MOR($H_2$), Ni-MOR(APE), Ag-ZSM5($H_2$), Ag-ZSM5(APE), Pd-Y($H_2$), Pd-Y(APE), Pd-MCM-22($H_2$), Pd-MCM-22(APE), Pd-β($H_2$), Pd-β(APE), Pt-MOR ($H_2$), Pt-MOR (APE), Au-MOR ($H_2$), and Au-MOR (APE) were similar with that of Cu-MOR ($H_2$) and Cu-MOR(APE) introduced above. The zeolite parents of MOR (Si/Al = 9), ZSM5 (Si/Al = 19), Y (Si/Al = 50), MCM-22 (Si/Al = 20), and β (Si/Al = 20) were all purchased from Tosho corporation in Japan. The sources of Fe, Co, Ni, Ag, Pd, Pt, and Au, were $Fe(NO_3)_3$·9$H_2$O (Wako, Japan), $Co(NO_3)_2$·6$H_2$O (Wako, Japan), $Ni(NO_3)_2$·6$H_2$O (Wako, Japan), $AgNO_3$ (Merck KGaA, Germany), $Pd(NO_3)_2$·2$H_2$O (Aladdin, China), $[Pt(NH_3)_4](NO_3)_2$ (Aladdin, China), and $AuCl_3$·3$H_2$O (Sinopharm, China), respectively.

The preparation process of Ag-ZSM(E-$NH_3$) was similar with that of Ag-ZSM5($H_2$) sample, but replacing the reducing gas by ammonia. The outside reduction was conducted in a reaction tube at 400 °C for 10 h with 5%$NH_3$ gas (95% Ar, 5% $NH_3$) of 40 mL/min.

Ag/$SiO_2$($H_2$) and Ag/$Al_2O_3$($H_2$) were prepared using an incipient wetness impregnation (IWI) method. Before the loading of silver, the $SiO_2$ (Fujisilysia, 75–150 μm, Japan) and $Al_2O_3$ (NST-7, Nikki-Universal Co., Ltd, Japan) carriers were pretreated in air at 450 °C for 3 h. Appropriate amounts of $AgNO_3$ (Merck KGaA, Germany) were dissolved in 3 g deionized water. The Ag precursor solution was finely mixed with 5 g carrier. The obtained humid solid was then degassed at vacuum for 48 h, and dried at 120 °C for 12 h before calcination at 500 °C for 3 h. Before catalytic evaluation, the samples were mildly reduced at 400 °C for 10 h with 40 mL/min $H_2$.

**Evaluation of catalytic performance.** DME carbonylation was carried out in a 9.5 mm-i.d stainless steel reactor, and the dosage of the catalyst was 0.5 g. Before being exposed to reactants, the catalyst was pretreated at 500 °C for 3 h in a flowing high-purity $N_2$ of 20 mL/min. The reaction took place under 1.5 MPa at a constant temperature of 220 °C, and the reaction gas of Ar/DME/CO (3.1% Ar, 4.1% DME, CO balance) was introduced into the reactor with a 20 mL/min flow rate. Gas-phase reaction products were analyzed by an online gas chromatograph with dual TCD detectors connected with Porapak Q and activated carbon columns, respectively. Liquid products collected by an ice trap were analyzed using another FID gas chromatograph with a connected capillary column (DB-624). The DME conversion ($C_{DME}$), products' carbon molar selectivities ($S_i$), and the space time yield of MA ($STY_{MA}$) were calculated with the following equations:

(1) $C_{DME}$ (%) = ($A_{DME,in}/A_{Ar,in} - A_{DME,out}/A_{Ar,out}$)/($A_{DME,in}/A_{Ar,in}$) × 100%
A is the GC peak area of detected species.

(2) $S_i$ (%) = $n_i/\sum n_i$ × 100%
i = MA, MeOH, $CO_2$, $CH_4$, acetic acid, n = the carbon molar number of i.

(3) $STY_{MA}$ (mmol/(kg h)) = ($n_{MA}/m_{Catalyst}$)/$t_{reaction}$ × 100%
$n_{MA}$ = the molar number of MA formed; $m_{Catalyst}$ = the mass of catalyst used in reaction; $t_{reaction}$ = the time of reaction.

Methane coupling reaction was conducted in a continuous fixed-bed reactor at 800 °C and atmospheric pressure. Typically, 0.5 g catalyst was loaded into a quartz reactor (inner diameter = 8.0 mm). Quartz wool plugs were employed to immobilize the catalyst bed. Before the reaction, the catalyst was heated to 800 °C (10 °C/min) in $N_2$ flow (30 mL/min). Then, a gas mixture of $CH_4$/Ar = 90.3%/9.7% (vol %) with flow rate of 12.5 mL/min was introduced into the reactor, to start the reaction. The effluent gas from the reactor to GC was heated on 200 °C, to prevent possible condensation of heavy hydrocarbon products. The effluent gas of Ar, CO, $CH_4$, and $CO_2$ was detected by an online GC with an activated carbon column and a TCD detector. The light hydrocarbons, such as methane, ethylene, and ethane, were analyzed by another online GC with a Porapak-Q column and an FID detector. The possible heavy hydrocarbons of benzene, toluene, naphthalene, were collected in an ice trap, and analyzed by an off-line GC with a capillary column (DB-1) and an FID detector. The methane conversion $C_{CH_4}$ and product selectivity $S_{C_xH_y}$ were calculated as follows:

(1) $C_{CH_4}$ (%) = ($A_{CH_4,in}/A_{Ar,in} - A_{CH_4,out}/A_{Ar,out}$)/($A_{CH_4,in}/A_{Ar,in}$) × 100%
A is the GC peak area of detected species.

(2) $S_{C_xH_y}$ (%) = ($x \times n_{C_xH_y}$)/($n_{CH_4,in} - n_{CH_4,out}$) × 100
x is the carbon number of $C_xH_y$; n is the mole number of detected species.

Methane oxidation reaction was accomplished in a stainless-steel autoclave. Typically, 10 mL deionized water solution of $H_2O_2$ (0.5 M) mixed with 10 or 30 mg catalyst was added into the autoclave. After sealing, the air in reactor was replaced for three times with 1 MPa reaction gas, then the reactor was fed with 90% $CH_4$ in Ar to 3 MPa. The motor was vigorously stirred at 1200 rpm, the temperature was raised to 50 or 70 °C to start the reaction at the same time. After the 30 min reaction, vessel was cooled by ice (<10 °C), to avoid volatilization of the products.

The products such as methyl hydroperoxide, methanol, and formic acid in the liquid phase were analyzed by NMR spectroscopy. $^1$H-NMR spectra were acquired on a JNM-ECX 400 spectrometer after filtration. The gas products were measured by a TCD-GC with Porapak Q column.

(1) MeOH selectivity (%) = MeOH yield (μmol)/$\sum$Product yield (μmol) × 100%

(2) MeOH productivity (μmol/(g h)) = μmol of $CH_3$OH in an hour/total weight of catalyst

**Structural characterization.** Powder XRD patterns of samples were collected using a Rigaku RINT 2400 X-ray Diffractometer equipped with a Cu-K (0.154 nm) X-ray source. The X-ray tube was operated at 40 kV and 40 mA. The variation of zeolite topological structure during the APE reduction was verified by in situ XRD (D8 Advance, Bruker). The XRD patterns were recorded at each 10 °C during the temperature range from 300 to 500 °C. The textural property of samples was determined by $N_2$ physisorption instrument (Quantachrome Nova 2200e). The morphologies of the samples characterized by high-resolution transmission electron microscope (TEM) were obtained with the JEOL JEM-2100 UHR TEM at 200 kV. In situ TEM was conducted by JEM-2100F: the temperature of H–Cu(3.27 wt%)-MOR($H_2$) sample was heated to 500 °C in the air with a rate of 10 °C/min. The TEM images were recorded at the fixed temperatures of 20, 100, 200, 300, 400, and 500 °C, respectively. When the temperature rose to 500 °C, it was kept for 40 min, and the images were recorded every 10 min. SEM images and energy-dispersive X-ray spectroscopy (EDS) elemental mappings were obtained via JSM-IT700HR at 20 kV and 40 mA. Metal amounts of samples were measured by X-ray fluorescence (XRF) using a Bruker S4 instrument at 60 kV and 150 mA. The BELCAT-M instrument equipped with a TCD was used to examine the reducibility of samples with the temperature-programmed reduction ($H_2$-TPR) method. The acidic properties of samples were tested with the temperature-programmed desorption of $NH_3$ ($NH_3$-TPD) using 50 mg powder at the temperature range of 100–700 °C. DTA of the prepared samples before APE reduction was performed on Shimadzu DTG-60 thermal analyzer. In a typical DTA measurement, the temperature was heated from room temperature to 850 °C with a ramping rate of 10 °C/min under air atmosphere. Mass spectrometer (MS) analysis of the gas-phase product of samples before APE reduction was conducted by QMS 403 Aëolos Quadro. In the online MS analysis, the solid sample was heated from room temperature to 600 °C with a ramping rate of 10 °C/min under Ar atmosphere. The generated gas-phase product was injected into the MS with Ar as the carrier gas. XPS and AES were conducted with a Thermo Fisher Scientific ESCALAB 250Xi instrument equipped with an Al-Kα X-ray radiation source ($hv$ = 1486.6 eV). Ar-ion sputtering was adopted to reveal the change of Cu species from the surfaces to the core of samples via increasing the etching time.

## Data availability
Source data are provided with this paper. The other data supporting the findings of this study are available from the corresponding authors on reasonable request. Source data are provided with this paper.

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

## Acknowledgements

This work was financially supported by the International Partnership Program (Grant No. 122214KYSB20170007) and the Key Research Program of Frontier Sciences (Grant No. QYZDB-SSW-JSC043) of Chinese Academy of Sciences. Financial aid from Ningxia Hui Autonomous Region Key R&D Program [No. 2019BFH02016] is greatly appreciated. The aid from Minghui Tan, Shengying Zhao, Hailun Geng, Bing Xu, and Xuemei Wu of Institute of Coal Chemistry, Chinese Academy of Sciences is greatly appreciated. Jie Yao

appreciates the Grant-in-Aid for JSPS Fellows. Finally, Jie Yao wants to thank, especially, the patience, care, and support from Wenjie Deng in the hardest time.

## Author contributions

J.Y., Y.H., and Y.Z. performed most of the experiments and analyzed the experimental data. X.F., J.F., and K.S. conducted the X-ray diffraction, scanning electron microscopy, and transmission electron microscopy, respectively. X.Y., W.Z., T.Z., and Z.G. performed the $N_2$ physisorption, X-ray fluorescence, temperature-programmed reduction, and differential thermal analysis, respectively. J.Y., X.P., and G.Y. wrote the first draft of the manuscript. X.P., G.Y., and N.T. designed the research, analyzed the data, and wrote the manuscript. All authors discussed the results and commented on the manuscript at all stages.

## Competing interests

The authors declare no competing interests.
