## [Peer Review File · Nature Communications]

Title: Ammonia Pools in Zeolites for Direct Fabrication of Catalytic CentersReviewers' comments:

Reviewer #1 (Remarks to the Author):

The authors have made some interesting observations and offer some plausible suggestions to interpret them. The work, with a large amount of improvement, could be publishable.

The gist of the matter is that ammonium form zeolites containing metal exchange ions (e.g., of Cu, Pd) can be reduced in-situ by ammonia formed by decomposition of the ammonium ions nearby. This process evidently proceeds more slowly than reduction by more conventional methods, such as with dihydrogen or ammonia gas flowing over the zeolite.

If developed thoroughly and well, this could be valuable for preparation of small metal clusters retained in the zeolite pores.

The authors have done some valuable characterization of their samples, but not enough. They have tested the catalysts for various reactions, but not thoroughly enough to demonstrate strongly the potential value of their method.

The characterization of the metals is not sufficient. The TEM images do not provide information about metals in the zeolite pores. Metals outside the pores, which are imaged, are potentially of much less interest than those inside the pores.

The TEM characterization would be more fruitful with heavier metals than the authors have used, because of the greater atomic numbers and better contrast.

Too much of the interpretation of the other characterization data (e.g., TPR) is assertion rather than inference based on appropriate comparisons/calibrations.

The language is overheated and inappropriate. Terms such as "dream reaction" and magic do not belong in a paper in this journal.

The way in which the authors introduce the catalytic results and characterization results is not organized very well. It would be better if the characterization data were presented first and used as a basis for interpreting the catalysis performance data.

Reviewer #2 (Remarks to the Author):

This manuscript presents a method to prepare Cu, Pd and Ag metal nanoparticles as supported inside zeolite micropores. The zeolites were ion-exchanged with NH_4^+ before supporting catalyst precursors. The NH_4^+ ion is decomposed to NH_3 upon heating, which acts as a reducing agent inside the zeolite pores. The authors claimed that the reduction of metal precursors in this manner has an advantage of the formation of micropore-hosted nanoparticle and accordingly high catalytic activity in some reactions. The authors attributed this phenomenon to the effect of in-situ ammonia generation, and referred it as 'ammonia pool effect' or APE in abbreviation. However, I do not agree that the APE is generally applicable to other catalyst systems. The formation of catalytic metal nanoparticles inside zeolite micropores depends not only reducing agents but also chemical nature of the metal precursors (ionic, covalent, coordination, monomeric, oligomeric...), zeolite cations, and the degree of zeolite dehydration. For instance, when Pt catalyst is supported on zeolite using platinum tetraammine nitrate

or hexachloroplatinic acid, the zeolite needs to be pretreated by calcination in oxygen or air before reduction. Large agglomeration of Pt particles occurs at external surfaces of the zeolite particles, instead of inside micropores. I am not saying that the APE is not good for Cu, Pd and Ag, but I think that the effect is difficult to generalize to other systems and experimental conditions. There are other reducing agents better than ammonia, depending on metals. For example, alcohol is a good reducing agent to support gold nanoparticles on zeolite. Even water vapor coming from dehydration of zeolite can be a reducing agent in the case of gold.

Overall, I do not recommend this paper for publication in NatureComm. This paper looks interesting and suitable for a catalysis journal, but it seems lacking general attention.

Response to Referees Letter

ID: NCOMMS-21-27264A-Z

Title: Ammonia Pools in Zeolites for Direct Fabrication of Catalytic Centers

Reviewer #1:

The authors have made some interesting observations and offer some plausible suggestions to interpret them. The work, with a large amount of improvement, could be publishable.

The gist of the matter is that ammonium form zeolites containing metal exchange ions (e.g., of Cu, Pd) can be reduced in-situ by ammonia formed by decomposition of the ammonium ions nearby. This process evidently proceeds more slowly than reduction by more conventional methods, such as with dihydrogen or ammonia gas flowing over the zeolite. If developed thoroughly and well, this could be valuable for preparation of small metal clusters retained in the zeolite pores.

We thank you for your positive and professional evaluation on our work, the suggestions from you can significantly improve the quality of our study. We answer your questions point-by-point as follows:

***Q1.** The authors have done some valuable characterization of their samples, but not enough. They have tested the catalysts for various reactions, but not thoroughly enough to demonstrate strongly the potential value of their method.*

According to your suggestion, we supplemented more valuable characterizations and found some new merits of our APE method. The added characterizations involved *in-situ* TEM, *in-situ* XRD, TG-DTA-MS, SEM-EDS elemental mapping, XPS, H₂-TPR, and TG. In addition, besides the as-proved metals like Cu, Ag, and Pd, more base metals (Fe, Co, Ni) and noble metals (Pt and Au) are also proved their easier reduction by our APE.

- We reconducted the TEM characterization of Cu-MOR and supplemented new

TEM images of Ag-ZSM5, Pd- β , and Pt-MOR, the corresponding results are
shown in the followed Fig. 1, Fig. 2, Fig. 3, and Fig. 4, respectively, as well as
in Fig. S9, Fig. S19, Fig. S20, and Fig. S21, respectively, in the new SI of the
revised manuscript. Based on the results given in Figs. 1-4, it is obvious that
the metal particles inside zeolite fabricated by APE are much smaller than
those by traditional H₂ reduction. In addition, the metal particles after H₂
reduction are easier to migrate to the outer surface of zeolite, but the metal
particles fabricated *via* APE are confined in the zeolite. This should be one of
the reasons why the size of metal particles generated by traditional H₂
reduction is larger than that by APE.

**Fig. 1.** The TEM images of **a**, Cu(1.14wt%)-MOR(APE); **b**, Cu(2.52wt%)-MOR(APE); **c**,
Cu(3.41wt%)-MOR(APE); **d**, Cu(3.27wt%)-MOR(H₂); and **e**, Cu(5.72wt%)-MOR(APE).

**Fig. 2.** The TEM images of **a**, Ag(2.4wt%)-ZSM5(APE) and **b**, Ag(2.3wt%)-ZSM5(H₂).

**Fig. 3.** The TEM images of **a**, Pd(1.3wt%)-β(APE) and **b**, Pd(1.3wt%)-β(H₂).

**Fig. 4.** The TEM images of **a**, Pt(0.9wt%)-MOR(APE) and **b**, Pt(0.9wt%)-MOR (H₂).

● We also supplemented the scanning electron microscope (SEM) images and
 energy-dispersive X-ray spectroscopy (EDS) elemental mappings analysis on
 the aluminum, silicon and metal of the surface of Cu-MOR, Ag-ZSM5, and

Pd-Y samples. The result is shown in the followed Fig. 5, Fig. 6, and Fig. 7,
respectively, as well as in Fig. S11, Fig. S22, and Fig. S23, respectively, in the
new SI of the revised manuscript. The elemental mappings of different
samples were obtained under the same accelerating voltage and collection
time. Evidently, for metal-zeolite (APE) samples, the abundance of metal
elements mapping is weaker than that of metal-zeolite (H₂) samples,
illustrating that there are more metal particles on the surface of metal-zeolite
(H₂) samples than that of metal-zeolite (APE) samples. This finding is
consistent with the result indicated by TEM in the above Figs. 1-4. Therefore,
we can conclude that our APE can effectively confine the reduction and
formation of smaller metals nano particles inside the zeolite, better than the
traditional H₂ reduction.

**Fig. 5.** The SEM-EDS elemental mappings of **a**, Cu(3.41wt%)-MOR(APE) and **b**, Cu(3.27wt%)-
MOR(H₂).

**Fig. 6.** The SEM-EDS elemental mappings of **a**, Ag(2.4wt%)-ZSM5(APE) and **b**, Ag(2.3wt%)-
 ZSM5(H₂).

**Fig. 7.** The SEM-EDS elemental mappings of **a**, Pd(1.4wt%)-Y(APE) and **b**, Pd(1.2wt%)-Y(H₂).

● We also utilized *in-situ* TEM to observe the precursor of H-Cu(3.27wt%)-
MOR(H₂) in air calcination, and the result is shown in the followed Fig. 8, as
well as in Fig. S10 in the new SI of the revised manuscript. The *in-situ* process
revealed rapid agglomeration of the copper species on the surface. These
results clearly demonstrated that traditional method of air calcination and H₂
reduction can easily lead to the agglomeration of copper particles, but APE
reduction is a successful way to inhibit it.

**Fig. 8.** *In-situ* TEM characterization of H-Cu(3.27wt%)-MOR(H₂) sample.

● Thermogravimetry-differential thermal analysis and mass spectrometry (TG-
DTA-MS) characterization was supplemented to *in-situ* trace the formation of
gas-phase product during APE reduction process. The differential thermal
variation and gas-phase product were collected and analyzed in real time by
DTA and MS, respectively. The result is displayed in the followed Fig. 9, as
well as in Fig. S4 in the new SI of the revised manuscript. As indicated in Fig.
9, N₂ is detected at the temperature range of 350-500 °C. It indicates that

ammonia is oxidized into N₂ molecule in this temperature range.
Correspondingly, copper species are reduced into their reduced state. This
characterization further proves the APE reduction process as proposed by us
in manuscript's equations.

**Fig. 9.** TG-DTA-MS characterization of NH₄-Cu(3.41wt%)-MOR sample.

● *In-situ* XRD characterization was also supplemented to further verify the
variation of zeolite topological structure during the APE reduction. The
pattern of *in-situ* XRD is showed in the followed Fig. 10, as well as in Fig. S8
in the new SI of the revised manuscript. The NH₄-Cu(3.41wt%)-MOR sample
was heated to 500 °C, and the XRD pattern was recorded in the temperature
range of 300-500 °C. Copper characteristic peak was not observed, because
of the high dispersity of copper particles. The framework of MOR zeolite was
also stable, certifying that both the high-temperature calcination and the
copper particles formation could not destroy the crystalline phase of MOR
zeolite.

**Fig. 10.** *In-situ* XRD characterization of NH₄-Cu(3.41 wt%)-MOR sample.

● Besides metals Cu, Ag, and Pd, more new base metals (Fe, Co, Ni) and noble
 metals (Pt and Au) were employed to fabricate metal-zeolite samples by APE
 or traditional H₂ reduction. The results of XRD, H₂-TPR, and XPS
 characterizations of these samples are listed in the followed Fig. 11, Fig. 12,
 and Fig. 13 (as well as in Fig. S15, Fig. S16, and Fig. S14 in the new SI of the
 revised manuscript). In Fig. 11, all samples have typical MOR characteristic
 peaks, and no metal diffraction peak is observed in all samples, implying that
 the MOR crystal phase structure of all samples is integrated, and the degree
 of metal dispersion is high. According to Figs. 12 and 13, H₂-TPR and XPS
 results both prove the superior reduction ability and wide applicability of APE,
 not only being suitable for base metals but also for noble metals. In all samples,
 more than 50% reduction degree of metal species can be obtained by APE
 method. It's worth mentioning here that zeolite dehydration occurs in the
 temperature-rise period of calcination. It is a fact that the water vapor coming
 from dehydration of zeolite can reduce Au. However, as proved by followed
 Fig. 12e, APE realizes a much higher reduction degree (92% vs 31%) than the
 water vapor reduction in the case of Au.

**Fig. 11.** XRD characterization of metal-MOR (metal=Fe, Co, Ni, Pt, and Au) samples fabricated by
 **a,** traditional H₂ reduction; and **b,** APE reduction, respectively.

**Fig. 12.** H₂-TPR characterization of metal-MOR (base metal = **a,** Fe; **b,** Co; and **c,** Ni; noble metal

= **d**, Pt and **e**, Au) samples with or without APE reduction. Note: Reduction degree = 1 - (the H₂
 consumption of APE sample / the H₂ consumption of the sample without reduction) (1 - (area of blue curve / area of
 red curve)). The Au(1.0wt%)-MOR(Ar) sample refers to the sample reduced by zeolite dehydration.

**Fig. 13.** The XPS profiles of **a**, Fe(1.8wt%)-MOR(APE); **b**, Co(0.9wt%)-MOR(APE); **c**,
 Ni(1.2wt%)-MOR(APE); **d**, Pt(0.9wt%)-MOR(APE); and **e**, Au(1.1wt%)-MOR(APE).

● The TG analysis of the used metal-zeolite catalysts fabricated by traditional
 H₂ reduction or APE reduction is also provided. The formed coke amount
 comparison of the used H-Cu(3.41wt%)-MOR(APE), used H-Cu(3.27wt%)-
 MOR(H₂), used Ag-ZSM5(APE) and used Ag-ZSM5(H₂) is displayed in Fig.
 14 and Fig. 15 here, as well as in Fig. S24 and Fig. S26 in the new SI of the

revised manuscript. The TG result in Fig. 14 illustrates that the used H-
 Cu(3.41wt%)-MOR(APE) has less coke amount than used H-Cu(3.27wt%)-
 MOR(H₂) catalyst, which proves a superior anti-carbon ability of the catalyst
 prepared via APE. Notably, the used Ag(2.4wt%)-ZSM5(APE) also represents
 much less coke amount than the used Ag(2.3wt%)-ZSM5(H₂), and the Ag-
 ZSM5 catalyst prepared by the traditional H₂ reduction formed a lot of hard
 coke in the methane coupling reaction occurring at 800 °C (see Fig. 15). These
 findings suggest that the superior anti-carbon ability of APE, not only works
 well on medium-temperature reaction (DME carbonylation, 220 °C) but also
 takes effect on high-temperature reaction (methane coupling, 800 °C).

**Fig. 14.** The comparison of TG profiles of used H-Cu(3.27wt%)-MOR(H₂) and used H-
 Cu(3.41wt%)-MOR(APE).

**Fig. 15.** The comparison of TG profiles of used Ag(2.3wt%)-ZSM5(H₂) and used Ag(2.4wt%)-
 ZSM5(APE).

● The TEM analysis and copper particle size distribution of fresh Cu(3.41wt%)-
MOR(APE) sample and collected Py-Cu(3.41wt%)-MOR(APE) sample after
100 h reaction are supplemented and shown in Fig. 16 here, as well as in Fig.
S25 in the new SI of the revised manuscript. The TEM patterns in the followed
Fig. 16 prove that after 100 h reaction at 220 °C, the size of copper
nanoparticle was considerably stable. This result implies a robust
agglomeration-resisting nature of the copper species obtained by APE.

**Fig. 16.** The TEM image and copper particle size distribution of **a**, fresh Cu(3.41wt%)-MOR(APE)
and **b**, collected Py-Cu(3.41wt%)-MOR(APE) after 100 h reaction.

*Q2. The characterization of the metals is not sufficient. The TEM images do not provide*
*information about metals in the zeolite pores. Metals outside the pores, which are*
*imaged, are potentially of much less interest than those inside the pores. The TEM*
*characterization would be more fruitful with heavier metals than the authors have used,*
*because of the greater atomic numbers and better contrast.*

The characterizations of metals such as SEM-EDS mappings, H₂-TPR, XPS, and
*in-situ* TEM have been supplemented, please refer to the answer to **Q1**. We also

reconducted the TEM characterization and supplemented the TEM images of metal-
zeolite samples with heavier metals (Pd- β , Ag-ZSM5, and Pt-MOR), and the result is
in Figs. 1-4 of **Q1**. As you suggested, nanoparticles of Pd, Ag, and Pt reveal better
contrast, and the metal particles fabricated by APE method are confirmed in the zeolite
pores. In contrast, the metal particles after H₂ reduction seem easier to migrate to the
outer surface of zeolite, and the size of metal particles generated by traditional H₂
reduction is larger than that by APE. The migration and agglomeration process of
copper particles to the zeolite outer surface was also confirmed on the traditional
method prepared H-Cu(3.27wt%)-MOR(H₂) by *in-situ* TEM characterization (Fig. 8 in
answer to **Q1**).

*Q3. Too much of the interpretation of the other characterization data (e.g., TPR) is*
*assertion rather than inference based on appropriate comparisons/calibrations.*

According to your suggestion, we have deleted the inappropriate assertion, and
conduct the inference based on appropriate comparisons and calibrations. For example,
the paragraph to interpret the H₂-TPR has been revised as follows:

‘Quantitative analysis of different copper species over the H-Cu(X)-MOR(APE)
samples was achieved by temperature programmed reduction of H₂ (H₂-TPR)
measurement. H-Cu(3.27wt%)-MOR(Air) and H-Cu(3.27wt%)-MOR(H₂) were
prepared from Cu²⁺ ion-exchange of H-MOR zeolite, and then calcined by air and
reduced through H₂, respectively. These two samples were also measured by the H₂-
TPR instrument. The H₂ consumption curves of the H-Cu(X)-MOR(APE), H-
Cu(3.27wt%)-MOR(Air) and H-Cu(3.27wt%)-MOR(H₂) were displayed in Fig. 2d, and
the corresponding peaks to different copper species were labeled based on the related
literatures ^{25, 26, 27, 28}. The H-Cu(3.27wt%)-MOR(H₂) did not show any characteristic
peak, because the copper species have been completely reduced. Compared with the H-
Cu(3.27wt%)-MOR(Air), all the H-Cu(X)-MOR(APE) exhibited lower temperature
and intensity of H₂ consumption peaks. It indicates that the copper species of the H-
Cu(X)-MOR(APE) was easier to reduce than that of H-Cu(3.27wt%)-MOR(Air).

Moreover, they have been partially reduced during the APE process. The percentages
of different copper species were calculated and displayed in Table S1. The total amount
of Brønsted acid sites, obtained from the TPD of NH₄-MOR (Fig. S6), was used to
determine the ion-exchange degree of copper. The results revealed that the proportion
of Cu⁰ in the H-Cu(X)-MOR(APE) increased with decreasing the copper exchange
degree (see Table S1).'

*Q4. The language is overheated and inappropriate. Terms such as "dream reaction"*
*and magic do not belong in a paper in this journal.*

The overheated and inappropriate terms have been deleted.

*Q5. The way in which the authors introduce the catalytic results and characterization*
*results is not organized very well. It would be better if the characterization data were*
*presented first and used as a basis for interpreting the catalysis performance data.*

We agree with your opinion, and the structure of the resubmitted manuscript has
been reorganized. The characterization data were presented first and then used as a basis
for interpreting the catalysis performance result.

**Reviewer #2:**

*This manuscript presents a method to prepare Cu, Pd and Ag metal nanoparticles as*
*supported inside zeolite micropores. The zeolites were ion-exchanged with NH₄⁺*
*before supporting catalyst precursors. The NH₄⁺ ion is decomposed to NH₃ upon*
*heating, which acts as a reducing agent inside the zeolite pores. The authors claimed*
*that the reduction of metal precursors in this manner has an advantage of the formation*
*of micropore-hosted nanoparticle and accordingly high catalytic activity in some*
*reactions.*

Thank you very much for your comments. The point-by-point responses to your
comments are shown as follows.

***Q1.** The authors attributed this phenomenon to the effect of in-situ ammonia generation,*
*and referred it as 'ammonia pool effect'; or APE in abbreviation. However, I do not*
*agree that the APE is generally applicable to other catalyst systems.*

In our work, we had employed five different zeolites and three metals to verify the
general applicability of APE. Three unsimilar reactions, whose reaction temperatures
and pressure were specifically distributed at 50-70°C and 3 MPa (methane oxidation),
220°C and 1.5-4 MPa (DME carbonylation), 800°C and 0.1 MPa (methane coupling)
respectively, were selected to prove the superiority of APE. All the results substantially
proved the general applicability and excellent capacity of APE.

To further prove the wide applicability of APE, as well as responding to your
doubts, besides metals Cu, Ag, and Pd, more new base metals (Fe, Co, Ni) and noble
metals (Pt and Au) were employed to fabricate metal-zeolite samples by APE or
traditional H₂ reduction. The results of XRD, H₂-TPR, and XPS characterizations of
these samples are listed in the followed Fig. 11, Fig. 12, and Fig. 13 (as well as in Fig.
S15, Fig. S16, and Fig. S14 in the new SI of the revised manuscript).

In Fig. 11, all samples have typical MOR characteristic peaks, and no metal
diffraction peak is observed in all samples, implying that the MOR crystal phase
structure of all samples is integrated, and the degree of metal dispersion is high.

The H₂-TPR analysis demonstrated that the reduction degree of the Fe(1.8wt%)-
 MOR(APE), Co(0.9wt%)-MOR(APE), and Ni(1.2wt%)-MOR(APE) reached 86%,
 51%, and 67%, respectively (Fig. 12a-c). Moreover, the reduction degree of the
 Pt(0.9wt%)-MOR(APE) and Au(1.1wt%)-MOR(APE) was as high as about 90% (Fig.
 12d and e). In control experiment, Au(1.1wt%)-MOR(Ar) was synthesized via
 calcination in Ar atmosphere, and *in-situ* reduced by zeolite dehydration. It is a fact that
 the water vapor coming from dehydration of zeolite can reduce Au. However, as proved
 by followed Fig. 12e, APE realizes a much higher reduction degree (92% vs 31%) than
 the water vapor reduction in the case of Au.

The XPS results revealed that, after APE reduction, all the Fe(1.8wt%)-
 MOR(APE), Co(0.9wt%)-MOR(APE), Ni(1.2wt%)-MOR(APE), Pt(0.9wt%)-
 MOR(APE) and Au(1.1wt%)-MOR(APE) exhibited characteristic peaks of the
 zerovalent metals (Fig. 13a-e). It clearly proved that these metal samples were reduced
 by the APE process. In particular, the Pt(0.9wt%)-MOR(APE) and Au(1.1wt%)-
 MOR(APE) generated higher proportion of zerovalent peaks, than the other samples.
 This indicates that the noble metals (Pt and Au) on the MOR were easier to be reduced
 than the transitional metals (Fe, Co, and Ni).

H₂-TPR and XPS results both prove the superior reduction ability and wide
 applicability of APE, not only being suitable for base metals but also for noble metals.

**Fig. 11.** XRD characterization of metal-MOR (metal=Fe, Co, Ni, Pt, and Au) samples fabricated by
 **a**, traditional H₂ reduction; and **b**, APE reduction, respectively.

**Fig. 12.** H₂-TPR characterization of metal-MOR (base metal = **a**, Fe; **b**, Co; and **c**, Ni; noble metal
 = **d**, Pt and **e**, Au) samples with or without APE reduction. Note: Reduction degree = 1 - (the H₂
 consumption of APE sample / the H₂ consumption of the sample without reduction) (1 - (area of blue curve / area of
 red curve)). The Au(1.0wt%)-MOR(Ar) sample refers to the sample reduced by zeolite dehydration.

**Fig. 13.** The XPS profiles of **a**, Fe(1.8wt%)-MOR(APE); **b**, Co(0.9wt%)-MOR(APE); **c**,
 Ni(1.2wt%)-MOR(APE); **d**, Pt(0.9wt%)-MOR(APE); and **e**, Au(1.1wt%)-MOR(APE).

*Q2. The formation of catalytic metal nanoparticles inside zeolite micropores depends*
 *not only reducing agents but also chemical nature of the metal precursors (ionic,*
 *covalent, coordination, monomeric, oligomeric;), zeolite cations, and the degree of*
 *zeolite dehydration.*

Thanks for your suggestion. We agree with that “The formation of catalytic metal
 nanoparticles inside zeolite micropores depends not only reducing agents, but also
 chemical nature of the metal precursors, zeolite cations, and the degree of zeolite
 dehydration”. However, these viewpoints are not in conflict with the novelty of our

work.

The metal-zeolite catalysts prepared by ion-exchange are very difficult to be
reduced directly. As you said, both calcination and reduction need high temperature and
are energy-consuming processes. However, our APE bypasses the energy-intensive pre-
calcination process and reduces the metals inside zeolite channels under air atmosphere
directly without using any additional reducing agents. Most importantly, as proved by
the above Figs. 12-13 in answer to **Q1**, APE works well not only on noble metals (Pt,
Au) but also on base metals (Fe, Co, Ni).

In addition, we also further proved the superiority of APE in encapsulating of
metal particles inside zeolite. We reconducted the TEM characterization of Cu-MOR
and supplemented new TEM images of Ag-ZSM5, Pd- β , and Pt-MOR, the
corresponding results are shown in the followed Fig. 1, Fig. 2, Fig. 3, and Fig. 4,
respectively, as well as in Fig. S9, Fig. S19, Fig. S20, and Fig. S21, respectively, in the
new SI of the revised manuscript. Based on the results given in Figs. 1-4, it is obvious
that the metal particles inside zeolite fabricated by APE are much smaller than those by
traditional H₂ reduction. In addition, the metal particles after H₂ reduction are easier to
migrate to the outer surface of zeolite, but the metal particles fabricated *via* APE are
confined in the zeolite. This should be one of the reasons why the size of metal particles
generated by traditional H₂ reduction is larger than that by APE.

**Fig. 1.** The TEM images of **a**, Cu(1.14wt%)-MOR(APE); **b**, Cu(2.52wt%)-MOR(APE); **c**,
 Cu(3.41wt%)-MOR(APE); **d**, Cu(3.27wt%)-MOR(H₂); and **e**, Cu(5.72wt%)-MOR(APE).

**Fig. 2.** The TEM images of **a**, Ag(2.4wt%)-ZSM5(APE) and **b**, Ag(2.3wt%)-ZSM5(H₂).

**Fig. 3.** The TEM images of **a**, Pd(1.3wt%)-β(APE) and **b**, Pd(1.3wt%)-β(H₂).

**Fig. 4.** The TEM images of **a**, Pt(0.9wt%)-MOR(APE) and **b**, Pt(0.9wt%)-MOR (H₂).

We also utilized *in-situ* TEM to observe the precursor of H-Cu(3.27wt%)-
 MOR(H₂) in air calcination, and the result is shown in the followed Fig. 8, as well as in
 Fig. S10 in the new SI of the revised manuscript. The *in-situ* process revealed rapid
 agglomeration of the copper species on the surface. These results clearly demonstrated
 that traditional method of air calcination and H₂ reduction can easily lead to the

agglomeration of copper particles, but APE reduction is a successful way to inhibit it.

**Fig. 8.** *In-situ* TEM characterization of H-Cu(3.27wt%)-MOR(H₂) sample.

Hence, it is a fact that we discovered and contributed a brand-new APE, by which
to reduce metal nanoparticles inside zeolite channels much better, more quickly and
more conveniently, overcoming a lot of existing problems faced by us, such as
calcination, high temperature, energy-consuming, sintering, etc.

More importantly, using in catalytic reactions, the catalysts fabricated by APE
demonstrated good anti-coking and anti-carbon deposition performance. The TG
analysis of the used metal-zeolite catalysts fabricated by traditional H₂ reduction or
APE reduction is provided. The formed coke amount comparison of the used H-
Cu(3.41wt%)-MOR(APE), used H-Cu(3.27wt%)-MOR(H₂), used Ag-ZSM5(APE) and
used Ag-ZSM5(H₂) is displayed in Fig. 14 and Fig. 15 here, as well as in Fig. S24 and
Fig. S26 in the new SI of the revised manuscript. The TG result in Fig. 14 illustrates
that the used H-Cu(3.41wt%)-MOR(APE) has less coke amount than used H-
Cu(3.27wt%)-MOR(H₂) catalyst, which proves a superior anti-carbon ability of the
catalyst prepared via APE. Notably, the used Ag(2.4wt%)-ZSM5(APE) also represents

much less coke amount than the used Ag(2.3wt%)-ZSM5(H₂), and the Ag-ZSM5
catalyst prepared by the traditional H₂ reduction formed a lot of hard coke in the
methane coupling reaction occurring at 800 °C (see Fig. 15). These findings suggest
that the superior anti-carbon ability of APE, not only works well on medium-
temperature reaction (DME carbonylation, 220 °C) but also takes effect on high-
temperature reaction (methane coupling, 800 °C).

**Fig. 14.** The comparison of TG profiles of used H-Cu(3.27wt%)-MOR(H₂) and used H-
Cu(3.41wt%)-MOR(APE).

**Fig. 15.** The comparison of TG profiles of used Ag(2.3wt%)-ZSM5(H₂) and used Ag(2.4wt%)-
ZSM5(APE).

The TEM analysis and copper particle size distribution of fresh Cu(3.41wt%)-
MOR(APE) sample and collected Py-Cu(3.41wt%)-MOR(APE) sample after 100 h
reaction are supplemented and shown in Fig. 16 here, as well as in Fig. S25 in the new
SI of the revised manuscript. The TEM patterns in the followed Fig. 16 prove that after
100 h reaction at 220 °C, the size of copper nanoparticle was considerably stable. This

result implies a robust agglomeration-resisting nature of the copper species obtained by
APE.

**Fig. 16.** The TEM image and copper particle size distribution of **a**, fresh Cu(3.41wt%)-MOR(APE)
and **b**, collected Py-Cu(3.41wt%)-MOR(APE) after 100 h reaction.

*Q3. For instance, when Pt catalyst is supported on zeolite using platinum tetraammine*
*nitrate or hexachloroplatinic acid, the zeolite needs to be pretreated by calcination in*
*oxygen or air before reduction. Large agglomeration of Pt particles occurs at external*
*surfaces of the zeolite particles, instead of inside micropores. I am not saying that the*
*APE is not good for Cu, Pd and Ag, but I think that the effect is difficult to generalize*
*to other systems and experimental conditions.*

Thank you very much for your suggestion. Here, we proved that our APE is
considerably efficient and powerful also to Pt-zeolite sample preparation. Based on
your recommendation, $[\text{Pt}(\text{NH}_3)_4](\text{NO}_3)_2$ was used as the source of Pt, and the Pt-MOR
sample was fabricated by our APE reduction, as well as the traditional H_2 reduction as
reference, respectively. The TEM result of these two samples is shown in the followed

Fig. 4 here and Fig. S21 in the new SI of the revised manuscript.

**Fig. 4.** The TEM images of **a**, Pt(0.9wt%)-MOR(APE) and **b**, Pt(0.9wt%)- MOR (H₂).

Clearly, being different from the Pt nanoparticles prepared by traditional H₂
reduction, the Pt nanoparticles obtained by APE are confined inside the micropores of
zeolite. In addition, the size of Pt nanoparticles obtained by APE is also smaller than
that by traditional H₂ reduction. Therefore, the results confirmed the general
applicability of our APE again. Moreover, the APE is also easy to generalize to other
metals. As in followed Figs. 2 and 3, it is obvious that the Ag and Pd particles inside
zeolite fabricated by APE are also much smaller than those by traditional H₂ reduction.

**Fig. 2.** The TEM images of **a**, Ag(2.4wt%)-ZSM5(APE) and **b**, Ag(2.3wt%)-ZSM5(H₂).

**Fig. 3.** The TEM images of **a**, Pd(1.3wt%)-β(APE) and **b**, Pd(1.3wt%)-β(H₂).

In addition, APE can also well reduce base metals (Fe, Co, Ni) and noble metals
 (Pt and Au). As in Figs. 12 in answer to **Q1**, the H₂-TPR analysis demonstrated that the
 reduction degree of the Fe(1.8wt%)-MOR(APE), Co(0.9wt%)-MOR(APE), and
 Ni(1.2wt%)-MOR(APE) reached 86%, 51%, and 67%, respectively (Fig. 12a-c).
 Moreover, the reduction degree of the Pt(0.9wt%)-MOR(APE) and Au(1.1wt%)-
 MOR(APE) was as high as about 90% (Fig. 12d and e).

The XPS results revealed that, after APE reduction, all the Fe(1.8wt%)-
 MOR(APE), Co(0.9wt%)-MOR(APE), Ni(1.2wt%)-MOR(APE), Pt(0.9wt%)-
 MOR(APE) and Au(1.1wt%)-MOR(APE) exhibited characteristic peaks of the
 zerovalent metals (Fig. 13a-e). It clearly proved that these metal samples were reduced
 by the APE process.

In summary, all results prove the superior reduction ability and wide applicability
 of APE.

*Q4. There are other reducing agents better than ammonia, depending on metals. For*

*example, alcohol is a good reducing agent to support gold nanoparticles on zeolite.*
*Even water vapor coming from dehydration of zeolite can a reducing agent in the case*
*of gold.*

Thank you very much for your comments. In this paper, we presented a new and
efficient APE for metal-zeolite catalysts preparation, which is not in conflict with the
known ways those you suggested.

Here, firstly, we want to emphasize that our work does not use ammonia to reduce
metal but find that NH₄-zeolite has potential self-reduction ability to the encapsulated
metals. Secondly, it is true that polyvinyl alcohol (PVA) can reduce Au nanoparticles in
some cases, as declared by this reviewer, but it is difficult to reduce more metals. In
addition, the employed expensive PVA and the co-produced wastewater by using this
method are potential problems too. In contrast, the APE in our work is greener, simpler,
more economical, and environmentally friendly. It provides one more choice for metal-
zeolite catalyst design, preparation, and application.

For the gold reduction, the water vapor coming from the dehydration of zeolite
can only reduce gold with lower reducibility. It is also difficult to reduce other metals
by using water vapor. To further identify the general applicability and excellent capacity
of APE, we also supply new evidence of TG-DTA-MS to check the process of zeolite
hydration of Au-MOR sample, and the result is displayed in the followed Fig. 17 (as
well as Fig. S17 in the new SI of the revised manuscript). Dehydration of zeolite during
the calcination process is clearly observed. However, the H₂-TPR result of Au-MOR
sample, as indicated in followed Fig. 12e (as well as Fig. S16e in the new SI of the
revised manuscript), proves that APE can realize the reduction degree of Au higher than
that of zeolite dehydration reduced sample (92% vs 31%).

Fig. 17. The TG-DTA-MS result of Au(1.1wt%)-MOR sample.

Fig. 12. H₂-TPR characterization of metal-MOR (base metal = a, Fe; b, Co; and c, Ni; noble metal

= **d**, Pt and **e**, Au) samples with or without APE reduction. Note: Reduction degree = 1 - (the H₂
consumption of APE sample / the H₂ consumption of the sample without reduction) (1 - (area of blue curve / area of
red curve)). The Au(1.0wt%)-MOR(Ar) sample refers to the sample reduced by zeolite dehydration.

REVIEWERS' COMMENTS

Reviewer #1 (Remarks to the Author):

This paper is much better than the original. The core idea, that ammonium ions in zeolites form ammonia that gently reduces metal cations in zeolites is satisfactorily demonstrated by the data. The presentation is more prudent than before, but it would still be a very good idea for the authors to pull back on statements such as that CO and H₂ are "explosive and poisonous." There is no insight there, and in practice safe work is done with these gases. Referring to catalysts as "more powerful" loose, overwrought language and should be removed. Even the term "ammonia pools" is inappropriate and should be avoided.

The evidence of the intermediate metal oxidation states is not very strong--other techniques, such as spectroscopic techniques, would provide a stronger foundation.

The catalyst preparation data are the interesting new results. The catalyst performance results are superficial and do not link strongly with the preparation data and would be best removed, leading to a shorter, sharper paper.

Overall, this work contains something new. It does not match up with the strongest papers in the journal, but it is as good as the weaker ones.

Reviewer #3 (Remarks to the Author):

In this paper titled by "Ammonia Pools in Zeolites for Direct Fabrication of Catalytic Centers", an original concept of 'Ammonia Pools in zeolites' was proposed. The named 'ammonia pool effect' (APE) is very interesting. Zeolites play important roles in the areas of chemistry and environment, but preparing metal-zeolite catalysts without post-treatments (such as erasing reduction by hydrogen under higher temperature), in a simpler and more environmentally friendly way, is still a challenge both in science and engineering. I believe that the APE contributed by this paper solved this problem in metal-zeolite catalysts preparation.

I checked all the materials submitted by authors, including the supporting information. This paper is well organized. The high-quality TEM images support the newly discovered encapsulation and confinement effects of APE to metallic nanoparticles. In comparison with traditional H₂ reduction, the APE in this paper is more powerful, feasible and scalable, being available to almost all zeolites and metals.

The authors also verified that the APE could encapsulate metallic Pt into zeolites successfully, where the traditional methods cannot realize in fact. Especially for the gold reduction, the gold reduction degree obtained by APE was 92%, much higher than the 31% by previously reported zeolite dehydration. In my opinion, the excellent in-situ reduction ability and broad application of APE for zeolites and noble/base metals are very promising, even in future plant engineering and commercialization. Therefore, I recommend this paper to be accepted for publication by Nature Communications after minor revision.

Few comments are listed below for authors:

In the section of Structural characterization, the recording method of in-situ TEM images is not clear. The patterns were recorded with the increase of temperature or at a fixed temperature?

A feed gas containing hydrogen was used to investigate the catalyst stability in carbonylation reaction. What is the purpose of hydrogen in feed gas?

Response to Referees Letter

ID: NCOMMS-21-27264A-Z

Title: Ammonia Pools in Zeolites for Direct Fabrication of Catalytic Centers

Reviewer #1:

This paper is much better than the original. The core idea, that ammonium ions in zeolites form ammonia that gently reduces metal cations in zeolites is satisfactorily demonstrated by the data.

We are very grateful to you for assessing our revised manuscript. We acknowledge the reviewers for your time and expertise. We are very appreciated to the reviewers for your constructive comments and suggestions. You help us to significantly improve our manuscript. Point-by-point responses to the remaining concerns of your comments are displayed as follows:

Q1. The presentation is more prudent than before, but it would still be a very good idea for the authors to pull back on statements such as that CO and H₂ are "explosive and poisonous." There is no insight there, and in practice safe work is done with these gases. Referring to catalysts as "more powerful" loose, overwrought language and should be removed. Even the term "ammonia pools" is inappropriate and should be avoided.

We are very thankful to you for your professional comment. According to your suggestion, the statements such as that CO and H₂ are ‘explosive and poisonous’ have been thoroughly pulled back in the revised manuscript. The loose, overwrought language such as ‘more powerful’ has been removed completely. However, for the term of ‘ammonia pools’, we think it will be better if it can be reserved, because this concept is the core of our work and can image the phenomena. This term is also considered to be an original concept in zeolites, and other reviewers also think that the ammonia pool

effect (APE) is very interesting and suitable.

Q2. The evidence of the intermediate metal oxidation states is not very strong, other techniques, such as spectroscopic techniques, would provide a stronger foundation.

We are very thankful to your professional suggestion. According to your comment, we supplemented another spectroscopic technique, Cu *LMM* auger electron spectroscopy (AES), to provide a stronger foundation and evidence of the intermediate metal oxidation states. The added characterizations are displayed in the Supplementary Fig. 5b and Fig. 5c and are also shown here as Fig. 1a and Fig. 1b as below.

Fig. 1 | a, The Cu *LMM* Auger spectra of H-Cu(X)-MOR(APE) samples obtained at 5 nm etching depth. **b**, Comparison of the Cu *LMM* Auger spectra on H-Cu(3.41wt%)-MOR(APE) and H-Cu(3.27wt%)-MOR(H₂).

The Cu *LMM* AES spectra of H-Cu(X)-MOR(APE) (See Supplementary Fig. 5b), as well as the comparison of Cu *LMM* AES spectra of H-Cu(3.41wt%)-MOR(APE) and H-Cu(3.27wt%)-MOR(H₂) (See Supplementary Fig. 5c) clearly prove the intermediate metal oxidation states of copper species in MOR zeolite. The spectra result certainly indicates that the increase of Cu content in MOR zeolite could lead to a higher Cu⁺ proportion but a lower Cu⁰ proportion in H-Cu(X)-MOR(APE) samples. And being compared with H-Cu(3.41wt%)-MOR(APE), the H-Cu(3.27wt%)-MOR(H₂) has a

higher Cu⁰ proportion. These findings agree well with the conclusion of H-TPR results in Fig. 2d of manuscript and the data in Supplementary Table 1.

Q3. The catalyst preparation data are the interesting new results. The catalyst performance results are superficial and do not link strongly with the preparation data and would be best removed, leading to a shorter, sharper paper.

We are very thankful to your professional suggestion. However, we think that the catalyst performance results are also the strong and convincing supports to the interesting new results of catalyst preparation. According to your suggestion, we have further simplified the text of catalyst performance, and made this work to a shorter and sharper paper. The main revised part of the catalyst performance in our manuscript is displayed as followed:

‘DME carbonylation is a core reaction in the new ethanol synthetic route proposed in our previous work ^{38, 39, 40, 41, 42, 43}. For zeolite-catalyzed DME carbonylation, it is gaining acceptance that reduced copper species can effectively promote the activity of zeolite catalysts ^{44, 45, 46, 47}. As in Supplementary Table S4, MOR zeolites were modified by copper species with different valence states. The metal copper (Cu⁰), via a traditional H₂ reduction, behaved the best catalytic performance. Oppositely, the monovalent and divalent copper species inhibited the carbonylation. Methanol as by-product was largely generated due to the hydration reaction of DME and H₂O ⁴⁸. This result agrees well with the reported conclusion and proves that Cu⁰ is a good promoter in zeolite-catalyzed DME carbonylation ^{45, 46}. ~~Anchoring Cu²⁺ ions in NH₄-MOR zeolites via simple ion exchange is an effective means of metal modification, and the copper species can be reduced via APE in situ. However, It~~ it is surprising that all the H-Cu(X)-MOR(APE) catalysts ~~displayed higher carbonylation performance displayed higher space-time yield (STY) of methyl acetate (MA),~~ than the H-Cu(X)-MOR(H₂) catalysts ~~in DME carbonylation~~ (Fig. 34a and Supplementary Table S5), ~~although the Cu⁰ content of~~ ~~Because the H-Cu(X)-MOR(APE) should possess lower Cu⁰ content than the H-Cu(X)-~~

~~MOR(H₂), as in Fig. 2d and Table S1. The lower Cu⁰ content was not in favor of DME carbonylation. In addition, the monovalent copper species on the Cu⁺ (3.71wt%) MOR also inhibited DME carbonylation (Table S4). H-Cu(X)-MOR(APE) was lower than that of H-Cu(X)-MOR(H₂) (Fig. 2d and Supplementary Table 1). These phenomena allow us to consider that the synergy of Cu⁰ and Cu⁺ may be the key factor in significantly promoting the DME carbonylation. Our characterization results also revealed that H-Cu(X)-MOR(APE) samples formed high ratio of Cu⁺ to Cu⁰ (Table S1). Moreover, In addition, the H-Cu(3.41wt%)-MOR(APE), with the highest ratio of Cu⁺ (40.6%) to Cu⁰ (45.8%) (See Supplementary Table 1), generated the best performance of DME carbonylation. ~~Thus~~ Hence, we confirm that the coexistence and synergy of Cu⁰ and Cu⁺ played the key role in enhancing the DME carbonylation of H-Cu(X)-MOR(APE).~~

Overall, this work contains something new. It does not match up with the strongest papers in the journal, but it is as good as the weaker ones.

We are very grateful to you for assessing our revised manuscript. We acknowledge the reviewers for your time and expertise. We very much appreciate the reviewers for your constructive comments and suggestions. You help us to significantly improve our manuscript.

Reviewer #3:

In this paper titled by 'Ammonia Pools in Zeolites for Direct Fabrication of Catalytic Centers', an original concept of 'Ammonia Pools in zeolites' was proposed. The named ammonia pool effect (APE) is very interesting. Zeolites play important roles in the areas of chemistry and environment, but preparing metal-zeolite catalysts without post-treatments (such as erasing reduction by hydrogen under higher temperature), in a simpler and more environmentally friendly way, is still a challenge both in science and engineering. I believe that the APE contributed by this paper solved this problem in

metal-zeolite catalysts preparation.

I checked all the materials submitted by authors, including the supporting information. This paper is well organized. The high-quality TEM images support the newly discovered encapsulation and confinement effects of APE to metallic nanoparticles. In comparison with traditional H₂ reduction, the APE in this paper is more powerful, feasible and scalable, being available to almost all zeolites and metals.

The authors also verified that the APE could encapsulate metallic Pt into zeolites successfully, where the traditional methods cannot realize in fact. Especially for the gold reduction, the gold reduction degree obtained by APE was 92%, much higher than the 31% by previously reported zeolite dehydration. In my opinion, the excellent in-situ reduction ability and broad application of APE for zeolites and noble/base metals are very promising, even in future plant engineering and commercialization. Therefore, I recommend this paper to be accepted for publication by Nature Communications after minor revision.

We are very grateful to you for assessing our revised manuscript. We acknowledge the reviewers for your time and expertise. We very much appreciate the reviewers for your constructive comments and suggestions. You help us to significantly improve our manuscript. Point-by-point responses to remaining concerns of your comments are displayed as follows:

Few comments are listed below for authors:

Q1. *In the section of Structural characterization, the recording method of in-situ TEM images is not clear. The patterns were recorded with the increase of temperature or at a fixed temperature?*

We are very thankful to you for the professional suggestion. According to your suggestion, we have supplemented the details of the recording method of in-situ TEM

images, the revised text is also displayed as follows:

‘*In-situ* TEM was conducted by JEM-2100F: the temperature of H-Cu(3.27wt%)-MOR(H₂) sample was heated to 500 °C in the air with a rate of 10 °C/min. The TEM images were recorded at the fixed temperatures of 20 °C, 100 °C, 200 °C, 300 °C, 400 °C, and 500 °C, respectively. When the temperature rose to 500 °C, it was kept for 40 min, and the images were recorded every 10 minutes.’

As shown, the patterns of the *in-situ* TEM were recorded at a fixed temperature.

Q2. A feed gas containing hydrogen was used to investigate the catalyst stability in carbonylation reaction. What is the purpose of hydrogen in feed gas?

We are very thankful to you for your professional question. Although the modification of pyridine is an effective measure to prepare a long-life MOR zeolite catalyst in DME carbonylation. However, with the increase of reaction duration, Py-MOR catalyst still behaves inevitable deactivation due to the carbon deposition derived from the side reaction of MTO in 12-MR of MOR zeolite. The addition of hydrogen in feed gas can effectively inhibit the deactivation of zeolite catalysts in the long-time running process, because the formation of coking precursor from the side reaction of MTO can be interdicted when there is hydrogen. We added hydrogen in feed gas to gain a long and stable catalytic performance of Py-Cu(3.41wt%)-MOR(APE) catalyst.